# Ablation of TrkB from Enkephalinergic Precursor-Derived Cerebellar Granule Cells Generates Ataxia

**DOI:** 10.3390/biology13080637

**Published:** 2024-08-20

**Authors:** Elena Eliseeva, Mohd Yaseen Malik, Liliana Minichiello

**Affiliations:** Department of Pharmacology, University of Oxford, Oxford OX1 3QT, UK; elena.eliseeva@pharm.ox.ac.uk (E.E.); yaseen.malik@pharm.ox.ac.uk (M.Y.M.)

**Keywords:** ataxia, cerebellar granule cells (GCs), TrkB, BDNF–TrkB signalling, enkephalinergic GC subset

## Abstract

**Simple Summary:**

Patients suffering from ataxia disorders show symptoms of incoordination often attributed to cerebellar dysfunction and eventual degeneration of Purkinje cells (PCs). In spinocerebellar ataxia 6 (SCA6), for example, reduced signalling of brain-derived neurotrophic factor (BDNF) through its receptor TrkB has been implicated in PC dysfunction and motor incoordination. Nonetheless, the TrkB receptor is also present in granule cells (GCs), which have extensive connections with PCs, suggesting that impaired BDNF–TrkB signalling in GCs would also negatively affect the function of PCs and possibly contribute to symptoms of motor incoordination in ataxia disorders. Here, we used a mouse model in which the TrkB receptor was removed from a specific subset of GCs to investigate whether this would induce motor incoordination in the mice. Analysis of these mice revealed a normal cerebellar structure and intact levels of selected synaptic markers. However, the mice exhibited motor incoordination symptoms and eventual PC dysfunction. Thus, dysfunctional BDNF–TrkB signalling in GCs alone was sufficient to induce symptoms of motor incoordination and may contribute to these symptoms in disorders such as SCA6. These findings can enhance our understanding of the causes of motor incoordination and help develop therapeutic interventions.

**Abstract:**

In ataxia disorders, motor incoordination (ataxia) is primarily linked to the dysfunction and degeneration of cerebellar Purkinje cells (PCs). In spinocerebellar ataxia 6 (SCA6), for example, decreased BDNF–TrkB signalling appears to contribute to PC dysfunction and ataxia. However, abnormal BDNF–TrkB signalling in granule cells (GCs) may contribute to PC dysfunction and incoordination in ataxia disorders, as TrkB receptors are also present in GCs that provide extensive input to PCs. This study investigated whether dysfunctional BDNF–TrkB signalling restricted to a specific subset of cerebellar GCs can generate ataxia in mice. To address this question, our research focused on *Trkb^Penk-KO^* mice, in which the TrkB receptor was removed from enkephalinergic precursor-derived cerebellar GCs. We found that deleting *Ntrk2*, encoding the TrkB receptor, eventually interfered with PC function, leading to ataxia symptoms in the *Trkb^Penk-KO^* mice without affecting their cerebellar morphology or levels of selected synaptic markers. These findings suggest that dysfunctional BDNF–TrkB signalling in a subset of cerebellar GCs alone is sufficient to trigger ataxia symptoms and may contribute to motor incoordination in disorders like SCA6.

## 1. Introduction

Ataxia, the major symptom of disorders such as spinocerebellar ataxias (SCAs), is usually caused by cerebellar dysfunction [1]. Predominantly cerebellar Purkinje cells (PCs) undergo degeneration in SCAs [2,3] and are dysfunctional in animal models of ataxia [4].

One of the signalling pathways implicated in ataxia is that of brain-derived neurotrophic factor (BDNF) through its high-affinity receptor TrkB. Specifically, it has been shown that BDNF mRNA is decreased in SCA6 patients’ cerebella [5] and Friedrich’s ataxia patient cells [6], and the BDNF protein is decreased in the SCA1 cerebellum [7]. Additionally, pharmacological (intraventricular) BDNF delivery had a therapeutic benefit for motor deficits and PC pathology at the early [8] and post-symptomatic [7,9] stages of the disease in different mouse models of SCA1.

Dysfunctional BDNF–TrkB signalling in PCs appeared to contribute to their dysfunction at an early disease stage in a mouse model of SCA6, since BDNF intensity was found to be reduced in PCs, and pre-onset treatment with a TrkB agonist improved the rotarod performance and PC firing frequency in these mice [10]. Interestingly, in the same study, the BDNF levels in the granule cell (GC) layer of the SCA6 mice were also decreased, and our exploration of a publicly available transcriptomic dataset from the cerebella of healthy humans and patients with ataxia telangiectasia (AT) [11] revealed that *NTRK2* expression was reduced in the GCs of the AT cerebella, suggesting that GC dysfunction may also contribute to cerebellar ataxias. This is supported by the fact that, according to transcriptomic studies, 69% of adult mouse GCs express *Ntrk2*, in contrast to only 27% of adult mouse PCs [12].

Furthermore, GC dysfunction would be expected to cause PC dysfunction. PCs receive extracerebellar input from two sources: directly from climbing fibres and indirectly from mossy fibres via the parallel fibres (PFs) of GCs, with each GC synapsing on 400 PCs and each PC receiving input from 55,000 GCs [13]. During cerebellar development, PF–PC synapses are important for the development of PC dendritic trees [14] and mediate the pruning of synapses between climbing fibres and PCs [15]. In adulthood, the silencing of GC outputs to PCs led to an increased regularity of spontaneous PC firing, impaired LTP and LTD induction at PF–PC synapses, and affected some forms of motor learning [16]. Therefore, dysfunctional BDNF–TrkB signalling in GCs could interfere with PC function and contribute to the ataxia symptoms in SCA6 and other ataxias where BDNF–TrkB signalling is abnormal. 

The fact that each GC synapses onto 400 PCs in the mouse cerebellum [13] raises the possibility that the dysfunction of a subset of GCs can propagate extensive dysfunction in PCs. Thus, the question is whether dysfunctional BDNF–TrkB signalling in a subpopulation of cerebellar GCs can generate ataxia in mice. 

To address this question, we used a mouse model, *Trkb^Penk-KO^* mice, in which the TrkB receptor is removed from a subset of cerebellar GCs, those derived from enkephalin-expressing GC precursors [17]. Enkephalin-like immunoreactivity is only found during the development of the rodent cerebellum [18,19]. Specifically, it appears that pre-proenkephalin (*Penk*) is expressed in GC precursors rather than differentiated GCs [20]. Moreover, according to publicly available transcriptomic datasets from developing mouse cerebella, only a subset of GC precursors express *Penk* [21]. Therefore, in *Trkb^Penk-KO^* mice, *Ntrk2* deletion occurs in GCs derived from enkephalinergic precursors.

In this study, we investigated the impact of deleting *Ntrk2* from enkephalin-expressing GC precursors. Specifically, we examined whether dysfunctional BDNF–TrkB signalling in a specific subset of cerebellar GCs is sufficient to cause ataxia in mice.

## 2. Materials and Methods

### 2.1. Animals

The animals were from mixed genetic backgrounds (C57BL/6J:129). Previously described *Trkb^Penk-KO^* mice [17] and their littermates (*Trkb^Penk-WT^*) were used in this study. These mice are a cross between an *Ntrk2* floxed line (*Trkb^lx/lx^*) [22] and a BAC-*Penk-Cre^tg/+^* strain, which carries Cre-recombinase under the control of the pre-proenkephalin (*Penk*) promoter [17]. The reporter line Rosa26-Ai9-tdTomato [23] was crossed with the BAC-*Penk-Cre^tg/+^* strain to generate BAC-*Penk-Cre^tg/+^*;Ai9 mice. The crossing of the BAC-*Penk-Cre^tg/+^*;Ai9 line with the *Trkb^lx/lx^* line produced *Trkb^Penk-KO^*;Ai9 mice. All experiments were conducted by an experimenter blind to genotype. 

The ages of the mice in this study ranged from P8 to 8 months. Both male and female mice were included in the different experiments. Same-sex littermates were group-housed in a temperature- and humidity-controlled vivarium. The mice had free access to food and water and were maintained on a 12h light/dark cycle (7:00–19:00).

### 2.2. Tissue Processing

After an intraperitoneal injection with pentobarbital (50 mg/kg), the mice were perfused with 0.1 M phosphate-buffered saline (PBS) followed by 4% paraformaldehyde, and their brains were harvested, embedded in an optimal cutting temperature (OCT) medium, and stored at −80 °C until use. 

The cerebella in the OCT blocks were sectioned sagittally with cryostat CM3030 S (Leica Biosystems, Nussloch, Germany). Three-month-old (3 M) cerebella were sectioned at 14 and 30 μm, 4 M cerebella were sectioned at 30 μm, P21 cerebella were sectioned at 30 μm, and P8 cerebella were sectioned at 20 and 30 μm. Additionally, the brainstem and spinal cord of a 3 M BAC-*Penk-Cre^tg/+^*;Ai9 mouse were sectioned coronally at 30 μm. Sections (14 μm) of 3 M cerebella were mounted directly on a microscope slide, and the rest of the tissues were suspended in 0.01% sodium azide in PBS at 4 °C. 

### 2.3. Immunofluorescence

Sections were washed two times with PBS at room temperature (10 min each), followed by incubation in blocking buffer (5% fish gelatine and 0.5% TritonX-100 in PBS) at room temperature for 1 h before overnight incubation with the primary antibodies diluted in the blocking buffer at 4 °C. The primary and secondary antibodies used in the immunofluorescence experiments are listed in Appendix A. After three washes with 0.1% TritonX-100 in PBS (10 min each), the sections were incubated for 2 hrs at room temperature with secondary antibodies diluted in the blocking buffer. After three washes with PBS, sections were incubated with 4′,6-diamidino-2-phenylindol (DAPI, 1:10,000 in PBS) at room temperature for 10 min, washed with PBS three times, and mounted in a Vectashield mounting media (Vector Laboratories, Burlingame, CA, USA). Fluorescent images were captured with a Leica DM6000B microscope (DFC365FX camera; Leica Microsystems, Wetzlar, Germany).

PENK immunostaining included an antigen retrieval step and followed the previously described protocol [24].

### 2.4. Estimation of a Proportion of GCs and PCs Undergoing Recombination 

The following cell-counting procedure was applied using Fiji [25] (ImageJ 2.14.0/1.54f, U.S. National Institutes of Health, Bethesda, MD, USA). NeuN-stained cerebellar sagittal sections (14 μm thick) were used. For GC estimation, three square regions of interest (ROIs) of 10,000 μm^2^ were randomly placed in the GC layer of three selected lobules per cerebellar section of a 3 M BAC-*Penk-Cre^tg/+^*;Ai9 mouse. NeuN-positive cells (GCs) were first counted in each ROI in the green channel, after which those NeuN-positive cells that were also tdTomato-positive (undergoing recombination) were counted in the red channel (Appendix A). For PC estimation, calbindin-stained cerebellar sagittal sections (14 μm thick) adjacent to those used for GC analysis were used. The total number of calbindin-positive cells (PCs) in the three selected lobules per cerebellar section was counted in the green channel, after which tdTomato-positive PCs were counted in the red channel. Six sections were analysed, three from sagittal level (SL) 11 of the Allen Brain Atlas (lobules analysed: culmen, simple lobule, and copula pyramidis; 1:2 sections) and three from SL13 (lobules analysed: culmen, declive, and pyramis; 1:2 sections). The proportion of GCs undergoing recombination was estimated by dividing the total number of NeuN-positive tdTomato-positive cells by the total number of NeuN-positive cells. Similarly, the proportion of PCs undergoing recombination required dividing the total number of calbindin-positive tdTomato-positive cells by the total number of calbindin-positive cells. 

### 2.5. PC Count Analysis

PC counting was performed using Fiji [25] (ImageJ, U.S. National Institutes of Health). Six evenly spaced calbindin-stained cerebellar sagittal sections (1 in 15, 30 μm thick, from SL9 to SL19 of the Allen Brain Atlas) per mouse from 4 M mice were analysed. The calbindin-positive PCs were counted in the simple lobule (at SL9 and 11), as well as lobules II (at SL15 and 17), III (at SL13, 15, and 17), IV–V (culmen; at SL9, 11, and 13), and X (nodulus; at SL15, 17, and19). The sums of these PCs (in lobule III, culmen, and nodulus, as well as the total in listed lobules) were then calculated for each mouse. 

### 2.6. Nissl Staining and Cavalieri Analysis

Sagittal (30 μm) cerebellar sections from 4 M and P21 animals were washed in PBS 3 times (10 min each), mounted onto glass slides, and dried overnight. The next day, they were fixed with 4% paraformaldehyde for 5 min, washed twice in PBS (5 min each), and defatted in xylene for 30 min. The sections were rehydrated in 100% ethanol for 1 min, 90% ethanol for 1 min, 70% ethanol for 40 s, 50% ethanol for 20 s, and stained in cresyl violet solution (0.02%) for 20 min. This was followed by a 10 s wash in PBS, dehydration in 50% ethanol (20 s), 70% ethanol (40 s), 90% ethanol (1 min), 100% ethanol (1 min), and clearing in xylene for 30 min. The sections were then embedded in Vectamount mounting media (Vector Laboratories). Brightfield images were captured with a Leica DM6000B microscope (DFC450C camera; Leica Microsystems).

Cerebellar volumes were estimated using the Cavalieri method with point counting on cresyl-violet-stained sections using Fiji [25] (ImageJ, U.S. National Institutes of Health). Thirteen evenly spaced cerebellar sections (P21, 1 in 7 sections; 4 M, 1 in 8 sections) per mouse from SL6 to SL18 of the Allen Brain Atlas were analysed. Each section was overlayed with a randomly positioned 10,000 μm^2^ regular grid, and the number of points (intersections of crosses) hitting the cerebellum was quantified in a blind manner. Volume was then computed according to the following formula: (1)Volume=Ap×ΣP×ssf×t,
where *Ap* stands for the area associated with each point (10,000 μm^2^), Σ*P* is the total number of points counted in all the sections, *ssf* is the section sampling fraction (7 for P21 and 8 for 4 M), and *t* is the mean section thickness (30 μm). 

To evaluate the reliability of the point density of the grids and sectioning intervals, the coefficient of error (CE) was calculated [26]. The shape factor of the whole cerebellum was set to 8.05, as calculated from the cerebellum of one P21 mouse. The accepted highest limit of CE is 5%, and the measured CE varied between 3.2 and 3.8%, meaning that it was always below the accepted limit.

The forebrain volumes of 1 M animals were estimated from coronal (30 μm) DAPI-stained forebrain sections. Four evenly spaced sections (1 in 14) per mouse from coronal level 51 to coronal level 69 of the Allen Brain Atlas were analysed. The area of each section was quantified using Fiji [25] (ImageJ, U.S. National Institutes of Health) and the volume of each forebrain was estimated using the following formula:(2)Volume=ΣA×ssf×t,
where Σ*A* is the sum of the areas quantified in all the sections, *ssf* is the section sampling fraction (14), and *t* is the mean section thickness (30 μm).

### 2.7. Immunoblotting

Western blots were performed on 3 animals per group. The mice were sacrificed by cervical dislocation after anaesthesia (intraperitoneal injection of pentobarbital, 50 mg/kg); their cerebella were dissected and snap-frozen. Tissues were homogenised in ice-cold RIPA buffer (150 mM NaCl, 1% TritonX-100, 0.5% sodium deoxycholate, 0.1% SDS, 50 mM Tris-HCl) containing protease and phosphatase inhibitors (S8830, SigmaFast, Thermo Scientific, St. Louis, MO, USA). After 5 min of sonication (30 s on/30 s off) and centrifugation (20 min at 20,000 rpm), the supernatants were collected, and the protein concentration of each sample was determined using a Pierce^TM^ BCA protein assay kit (23227, Thermo Scientific). The lysates were treated with loading buffer (LI-COR protein loading buffer, LI-COR Biosciences, Lincoln, NE, USA) and 100 mM DL-Dithiothreitol (DTT) at 95 °C for 10 min. The total protein lysates were then resolved on SDS-PAGE gels and transferred to nitrocellulose membranes. These membranes were blocked for 1 h at room temperature in blocking buffer (BB; 2% fish gelatine, 0.2% Tween-20 in PBS) and incubated overnight at 4 °C with primary antibodies in antibody incubation buffer (AIB; 1% fish gelatine and 0.1% Tween-20 in PBS). The primary and secondary antibodies used in the immunoblotting experiments, the corresponding gel percentages, and the amounts of proteins loaded are listed in Appendix A. The membranes were washed three times for 5 min in PBS with 0.1% Tween-20 (PBS-T) and incubated for 1 h at room temperature with secondary antibodies diluted in AIB. The membranes were washed three times for 5 min in PBS-T and the blots were visualised with the Odyssey M Imaging system (LI-COR Biosciences, Lincoln, NE, USA). Tubulin was used as the loading control. Immunoreactive bands were analysed using the Empiria Studio software version 2.1.0.134 (LI-COR Biosciences). 

### 2.8. Behavioural Analysis

#### 2.8.1. Hindlimb Clasping

Hindlimb clasping has been used as a marker of the severity of motor dysfunction [27]. It was assessed as described previously [28]. Mice were suspended by their tails and the hindlimb position was observed for 10 s to determine the extent and duration of hindlimb retraction. A score of 0 was given if the hindlimbs were splayed outward and never retracted towards the abdomen. If one hindlimb retracted towards the abdomen for more than 5 s, the mouse was assigned a score of 1. If both hindlimbs partially retracted towards the abdomen for more than 5 s, the mouse received a score of 2. Finally, if the hindlimbs entirely retracted towards the abdomen for more than 5 s, a score of 3 was given. Each mouse was tested twice. 

The test was performed on 7 mice (3 controls and 4 mutants; age range 7–8 months) and 18 mice (9 controls and 9 mutants; age range 3–4 months).

#### 2.8.2. Kyphosis Test

The dorsal curvature of the spine was assessed [28]. The mice were removed from their cages, placed on a flat surface, and observed as they walked. A score of 0 was given if the mouse had no kyphosis and could straighten its back as it walked. A score of 1 corresponded to mild kyphosis when stationary, but no kyphosis when the mouse walked. A score of 2 was assigned if the mouse maintained mild but persistent kyphosis, even when walking. Finally, a score of 3 was assigned if the mouse exhibited pronounced kyphosis as it walked and while stationary. 

The test was performed on 18 (6 controls and 12 mutants; age range 7–9 months) and 19 (10 controls and 9 mutants; age range 3–4 months) mice.

#### 2.8.3. Ledge Test

A ledge test was carried out [28] to assess motor coordination. The mice were placed on the ledge of a cage and monitored for a minute. A blinded observer scored their ability to walk along the ledge on a scale of 0–3. A score of 0 was assigned if the mouse generally did not lose balance but exhibited non-consecutive slips while turning the corners of the ledge (corner slips). If the mouse experienced single-paw slips or transient crawling, a score of 1 was given. A score of 2 was assigned if the mouse had a total of one or two double-paw slips or major corner slips. Major corner slips were defined as slips where the mouse did not immediately regain its balance or, immediately after regaining balance, slipped again. When a mouse had more than two double-paw slips or major corner slips, fell or nearly fell off the ledge, crawled for a prolonged period, or remained immobile for more than 20 s accompanied by tremor, it received a score of 3. 

The scoring was performed on 25 mice (9 controls and 16 mutants; age range 7–9 months) and 25 mice (13 controls and 12 mutants; age range 3–4 months), and these included mice that underwent the hindpaw clasping and kyphosis tests described in the previous sections. 

#### 2.8.4. Catwalk Test

An automated gait analysis was performed using the CatWalk System (Noldus Information Technology, Wageningen, The Netherlands). The apparatus consisted of an enclosed walkway on an LED-illuminated glass plate with a camera situated under the glass to record the illuminated footprints. The mice were individually placed at one end of the walkway and filmed freely crossing to the other end. A minimum of four compliant consecutive runs per mouse were recorded. Compliant runs were defined as those with a minimum of eight consecutive steps per run and a maximum allowed speed variation of 70%. The runs were checked, classified, and analysed using the CatWalk XT 10.6 software (Noldus Information Technology). The gait parameters were averaged for all runs. Only kinetic, temporal, and interlimb coordination parameters [29,30] were analysed (see Appendix A for the full list of parameters).

In total, 14 mice (5 controls and 9 mutants, age range 3–5 months) were analysed.

### 2.9. Statistical Analysis

Publicly available single-cell and single-nucleus RNA-Seq datasets [11,12,21,31,32] were accessed at https://singlecell.broadinstitute.org/single_cell/study/SCP1300/ (accessed on 10 June 2024), https://singlecell.broadinstitute.org/single_cell/study/SCP795/ (accessed on 25 September 2023), https://cellseek.stjude.org/cerebellum/ (accessed on 12 August 2023), http://cotneyweb.cam.uchc.edu/E10_E17_shinyCell/ (accessed on 22 August 2023), and http://allexpsctl.spinalcordatlas.org/ (accessed on 26 June 2024). To generate the overlayed *Penk* and *Ntrk2* expression images seen in Figure 1D,I, images of *Penk* expression (Figure 1B,F) and *Ntrk2* (Figure 1C,H) expression were superimposed using Adobe Photoshop (v.2023). Photoshop was also used to generate the coloured borders around selected subpopulations, as seen in Figure 1F–I. 

All statistical analyses were conducted using GraphPad Prism (version 9, GraphPad Software, San Diego, CA, USA). 

The cerebellar and forebrain volumes, PC counts in lobule III, the culmen, and the nodulus, and the total PC counts were analysed using two-tailed unpaired *t*-tests. Additionally, mouse weights at P21, P29, 3 M, and 8 M were compared with multiple two-tailed unpaired *t*-tests. 

The Western blot analysis results were first analysed using the Shapiro–Wilk test to determine the normality of residuals. An F-test of the equality of variances was then carried out. Then, if the residuals were normal and the equality of variances assumption was not violated, the data were analysed using a two-tailed unpaired *t*-test. Otherwise, the data were analysed either using Welch’s *t*-test (if the equality of variances assumption was violated but the residuals were normal) or using the Mann–Whitney test (if the residuals were not normal). For a Western blot analysis of the TrkB levels, the data were averaged across two gels. Values are presented as means ± SEM.

The ledge scores were analysed using multiple Mann–Whitney tests followed by Bonferroni–Dunn corrections for multiple comparisons. Behavioural data are shown as whisker-box plots with medians (whiskers: 5th to 95th percentile, box: 25th to 75th percentiles). The gait parameters were tested for normality with the Shapiro–Wilk test and the equality of variances was assessed by Levene’s test. Parameters that failed the normality assumption were analysed using the Wilcoxon signed-rank test, whereas the rest of the parameters were analysed using Welch’s *t*-test. The resulting *p*-values were corrected for multiple comparisons using the Holm–Bonferroni method. The weights of animals assessed in the CatWalk test did not differ between genotypes: mean weight of controls = 44.60 ± 6.94 g, mean weight of mutants = 49.31 ± 3.82 g, *p* = 0.139. Therefore, weights were not included in the analysis. 

*p* < 0.05 was considered to be statistically significant. 

## 3. Results

### 3.1. Exploring the Impact of Ntrk2 Deletion in a Specific Subset of Cerebellar Granule Cells

Decreased cerebellar BDNF–TrkB signalling is a feature of some cerebellar ataxias such as SCA6, Friedreich’s Ataxia, and SCA1 [5,6,7], which includes evidence of decreased BDNF levels in the GC layer of SCA6 mice [10]. In addition, by exploring a publicly available transcriptomic dataset from the cerebella of healthy humans and patients with ataxia–telangiectasia (AT) [11], we observed that *NTRK2* expression was reduced in the GCs of AT cerebella, but not in PCs (Figure 2A–C). In contrast, the expression of *BDNF* was not significantly affected in the GCs of AT cerebella (Figure 2A,D,E). This suggests that BDNF–TrkB signalling is specifically reduced in AT GCs. Thus, BDNF–TrkB signalling is reduced in cerebellar GCs in certain ataxia disorders, potentially contributing to ataxia symptoms.

To investigate whether abnormal BDNF–TrkB signalling in a subset of GCs is sufficient to evoke ataxia, we used *Trkb^Penk-KO^* mice carrying *Ntrk2* deletion in their enkephalinergic precursor cells. This study required first examining the expressions of *Penk* and *Ntrk2* in developing and adult mouse cerebella using previously published transcriptomic datasets [12,21,31], followed by a detailed analysis of the recombination pattern of the BAC-*Penk-Cre* cerebellum. 

*Penk* expression in mouse cerebella between E10 and P10 is relatively low based on a publicly available scRNA-Seq dataset [21]. Most of the expression was observed in the cluster of cells identified as GCs and their precursors, while some sporadic expression was also noted in different cell types, including glutamatergic deep cerebellar nuclei, GABAergic progenitors, and *Atoh1*-expressing progenitors (Figure 1A,B). Of note is that *Penk* was present predominantly in the clusters of GCs and their precursors defined by the presence of the glutamatergic progenitor marker *Atoh1* (Figure 1E–G), a marker that is downregulated by GC precursors when they exit the cell cycle [33]. Therefore, *Penk* is primarily expressed in GC precursors.

*Ntrk2* expression in the mouse cerebella between E10 and P10 was relatively high and appeared in most cerebellar cell types, with a lower expression in the glutamatergic deep cerebellar nuclei and glia (Figure 1C,H). The overlay of *Penk* and *Ntrk2* expression revealed the most co-expression in the clusters identified as GCs and their precursors (Figure 1D). Specifically, co-expression was observed in clusters 11, 2, 7, and 4 of the GCs and their precursors (Figure 1I), where the *Atoh1* marker was still present. Therefore, in *Trkb^Penk-KO^* mice, *Ntrk2* deletion would be expected to take place in enkephalinergic GC precursors, with specific subpopulations of GCs expected to already become dysfunctional during cerebellar development. 

Exploration of the re-analysed [31] scRNA-Seq dataset [21] of E10 and E17 mouse cerebella allowed us to investigate the origin of enkephalinergic *Ntrk2*-expressing GCs in greater detail. We used *Atoh1* as a marker of excitatory progenitors and *Pax6* as a marker of proliferative GCs. In total, 1.75% of *Atoh1*-expressing and 1.29% of *Pax6*-expressing sampled cells expressed enkephalin between E10 and E17 (Table 1). Therefore, in *Trkb^Penk-KO^* mice, GCs from which *Ntrk2* was deleted would be expected to descend from enkephalinergic precursors in embryonic stages, with a minimum of 1.29% of GCs affected. *Ntrk2* was expressed in 12.63% of *Atoh1*-expressing cells and 21.42% of *Pax6*-expressing cells sampled between E10 and E17 (Table 1); 14.79% of the enkephalinergic excitatory progenitors, GC precursors, and GCs sampled between E10 and E17 expressed *Ntrk2* (Table 1). Thus, a proportion (14.79%) of the cells in the GC lineage undergoing *Ntrk2* deletion in the embryonic cerebellum of *Trkb^Penk-KO^* mice would be expected to be dysfunctional. This suggests that the embryonic development of the *Trkb^Penk-KO^* cerebellum may be abnormal. 

Additionally, an analysis of the expression profile of *Penk* transcripts using a public portal of single-cell transcriptome profiling of adult (P90) mouse cerebella [12] revealed that *Penk* expression was low in adult GCs, corresponding to only 1.03% of the sampled cerebellar GCs (Table 2, Figure 1J,K,M), whereas *Ntrk2* was expressed in 69.47% of the sampled adult GCs (Table 2, Figure 1L,M). On the other hand, *Penk* and *Ntrk2* were co-expressed in adult Golgi cells, with *Penk* expressed in 81.35% of Golgi cells (Table 2, Figure 1K,M) and *Ntrk2* expressed in 99% of Golgi cells (Table 2, Figure 1L,M). 

Altogether, with the help of transcriptomic data, we were able to identify the cerebellar cells that are most likely to be affected in *Trkb^Penk-KO^* mice. Specifically, a subset of GCs would be expected to undergo Cre-recombination during development. In contrast, most Golgi cells would be expected to undergo Cre-recombination in adulthood in *Trkb^Penk-KO^* mice.

### 3.2. Pattern of BAC-Penk-Cre Mediated Recombination in the Cerebellum

To study the cerebellar recombination pattern of the BAC-*Penk-Cre* mouse line, we crossed it with the Rosa-Ai9-tdTomato line [23], which expresses the red fluorescent protein tdTomato upon Cre-mediated recombination. In the cerebellum of adult (3 M) BAC-*Penk-Cre^tg/+^*;Ai9 mice, tdTomato was primarily found in the GCs, with some recombination observed in the interneurons of the molecular layer, in a few PCs (Figure 3A) and in relatively few cells of the deep cerebellar nuclei (Appendix A). Immunostaining with NeuN, a specific marker for mature GCs in the cerebellum [34], revealed that only a subset of GCs expressed tdTomato (Figure 3B). Specifically, we found that approximately 37% of GCs expressed tdTomato in the cerebella of BAC-*Penk-Cre^tg/+^*;Ai9 mice. Since *Ntrk2* expression is found in about 69% of adult GCs in the mouse cerebellum [12], approximately 26% of GCs would be expected to become dysfunctional upon *Ntrk2* deletion (assuming that *Ntrk2* distribution is the same among tdTomato+ and tdTomato− cells). Immunostaining with calbindin revealed that only a few PCs were tdTomato+ (Figure 3C).

PENK immunostaining revealed that, while some cells in the cerebellar GC layer, presumably Golgi cells, were enkephalinergic, GCs did not express PENK at this stage (Figure 3D). Therefore, the Cre-recombination in GCs must have occurred during the development of the *Trkb^Penk-KO^* mice, rather than in adulthood. These data are consistent with transcriptomic data from adult mouse cerebella [12], showing that very few (1%) GCs express enkephalin, whereas the majority of Golgi cells (81%) do. Notably, Golgi cells were not tdTomato+ in the BAC-*Penk-Cre^tg/+^*;Ai9 mice, suggesting they would not undergo Cre-recombination in *Trkb^Penk-KO^* mice, despite their enkephalinergic status. 

In the P8 cerebellar primordium, tdTomato expression was already observed in the external granular (EGL) and internal granular (IGL) layers, two zones containing GC lineages at this stage (Appendix A), with only a subset of mature GCs being tdTomato+ (Appendix A). However, no PENK expression was observed in the GCs in either the EGL or IGL (Appendix A), suggesting that the tdTomato+ GCs must have previously expressed enkephalin and were descendants of enkephalinergic precursors. This subset of GCs will be referred to as Enk-derived GCs.

In the brainstem, tdTomato was expressed in the cochlear nucleus complex, principal sensory and spinal nuclei of the trigeminal nerve, and the nucleus of the solitary tract (Appendix A). Neither of these nuclei is implicated in motor function. In the spinal cord, scattered tdTomato expression was observed in the dorsal horn (Appendix A), consistent with previous research on enkephalinergic cell types in the spinal cord [35,36]. However, previous studies [37], as well as our exploration of publicly available transcriptomic datasets of adult mouse spinal cords [32,38] (Appendix A), show that the co-expression of *Penk* and *Ntrk2* is not high in the spinal cord, which, combined with the fact that relatively few cells in the spinal cord of a BAC-*Penk-Cre^tg/+^*;Ai9 mouse underwent recombination, suggests that it is unlikely that the spinal cord function of *Trkb^Penk-KO^* mice is affected.

### 3.3. Unaltered Cerebellar Morphology of Trkb^Penk-KO^ Mice

As a consequence of *Ntrk2* deletion from Enk-derived GCs, levels of full-length TrkB protein were reduced in 3 M *Trkb^Penk-KO^* cerebella (Appendix A).

To determine the impact of TrkB ablation from Enk-derived GCs on the cerebellum, we first examined the cerebellar anatomy in P21 and 4-month-old mice. There was no apparent gross abnormality in the number or appearance of the lobules and cortical layers of Nissl-stained *Trkb^Penk-KO^* cerebella, neither at P21 nor at 4 M (Figure 4A,B). Cavalieri analysis of the cerebellar volumes of P21 and 4 M *Trkb^Penk-WT^* and *Trkb^Penk-KO^* mice revealed that mutant cerebella were significantly smaller at P21, but not at 4 M (Figure 4C), thus suggesting that, while cerebellar development is delayed in *Trkb^Penk-KO^* mice, mutant cerebella eventually reach maturity. However, the *Trkb^Penk-KO^* mice were not significantly smaller than their littermate controls at P21, P29, 3 M, or 8 M (Appendix A), and the forebrain volumes of the mutants at 1 M were not significantly different from those of their littermate controls (Figure 4D). 

The pattern of tdTomato expression was largely similar in the P21 and 4 M BAC-*Penk-Cre^tg/+^*;Ai9 and *Trkb^Penk-KO^*;Ai9 mice (Figure 4E,F), suggesting that *Ntrk2* deletion did not significantly affect the overall structure of the cerebellum.

Since cerebellar GCs provide input to PCs, we wanted to explore whether the PCs would become dysfunctional in the *Trkb^Penk-KO^* cerebella due to *Ntrk2* deletion from Enk-derived GCs. Immunohistochemical staining of 3 M cerebella with calbindin did not reveal any gross abnormalities in the dendritic trees of the PCs in mutants (Figure 4G). Additionally, there was no difference in the total PC counts between the controls and mutants (Figure 4H) nor the counts within specific lobules, including lobule III, the culmen, or the nodulus (Appendix A). 

### 3.4. Molecular Consequences of Ntrk2 Deletion in Trkb^Penk-KO^ Mice

To further explore whether PCs are affected by *Ntrk2* deletion in Enk-derived GCs, we measured the cerebellar calbindin levels in 3 M *Trkb^Penk-KO^* mice and controls through Western blotting. The calbindin levels were not changed (Figure 5A). 

To explore whether PCs are affected by age, we compared cerebellar calbindin levels in 8 M mice. Interestingly, we found the calbindin levels significantly decreased by 23% in the 8 M *Trkb^Penk-KO^* cerebella (Figure 5B). Therefore, PCs may be either lost or functionally affected in 8 M *Trkb^Penk-KO^* mice.

Previous studies have shown that cerebellar synaptic development and maintenance depend on intact BDNF–TrkB signalling [39,40]. Therefore, to begin understanding how the cerebellar function may be affected in *Trkb^Penk-KO^* mice, we explored the cerebellar synaptic function in 3-month- and 8-month-old mice by testing for molecular changes in their synaptic proteins using Western blotting. We found that synaptophysin levels, a major integral membrane protein of secretory vesicles, were not significantly different in the cerebella of the *Trkb^Penk-KO^* mice from those of their littermates at 3 M (Figure 5A). Neither PSD95 nor GAD67 levels were affected at this stage (Figure 5A). Similarly, we observed that the synaptophysin, GAD67, and PSD95 levels remained unchanged in 8 M *Trkb^Penk-KO^* cerebella (Figure 5B). Therefore, based on these results, there was no evidence to suggest specific cerebellar synaptic molecular changes in the *Trkb^Penk-KO^* mice at 3 M and 8 M.

### 3.5. Behavioural Consequences of Ntrk2 Deletion in Enk-Derived GCs

Since the cerebellum is involved in motor function, we investigated the impact of Enk-derived GC dysfunction on this process. To this end, we tested 3 M and 8 M *Trkb^Penk-KO^* mice and their littermate controls with some tests described in the simple composite phenotype scoring system [28]: hindlimb clasping, kyphosis, and the ledge test—along with CatWalk gait analysis.

Consistent with the previous findings [17], neither the 3 M nor 8 M *Trkb^Penk-KO^* mice presented with hindlimb clasping (Table 3). Similarly, in the kyphosis test, all but one mouse scored 0, with one 8 M mutant achieving a score of 2 (persistent but mild kyphosis) (Table 3). However, the results of the ledge test (Figure 6A) revealed impaired motor coordination in the mutants at 3 M and 8 M (Figure 6B). While it appears that the mice performed worse with age, this difference was not significant either for the controls or mutants (controls: Mann–Whitney *U* = 34.00, adjusted *p* = 0.192; mutants: Mann–Whitney *U* = 67.00, adjusted *p* = 0.322). These results suggest that the mutants were significantly impaired from an early stage (3 M). For the representative videos of the ledge performances of the 3 M control and mutant mice, see Appendix A. 

The findings of the ledge test support its sensitivity and usefulness in detecting mild motor coordination impairment, which was not apparent when we previously used the rotarod test at 3 M and 8 M in *Trkb^Penk-KO^* mice [17]. 

To further understand their coordination deficit, we analysed the gaits of 3–5-month-old *Trkb^Penk-KO^* mice using the CatWalk system. Five-month-old *Trkb^Penk-KO^* mice and their littermates have been previously tested on the CatWalk [17], however, we focused our research on investigating the parameter groups previously shown to be affected in the cerebellar ataxias in humans [41] and included 3-month-old mice in the test, as this is the age of onset of ledge deficit in *Trkb^Penk-KO^* mice. The CatWalk test showed significant differences between genotypes in two of the gait parameters evaluated. There was a significant backward shift in the position of the left hindpaw relative to the ipsilateral frontpaw in the *Trkb^Penk-KO^* mice compared to the controls (Figure 6C,D). Additionally, the mean swing duration of the left hindpaw was significantly shorter in the *Trkb^Penk-KO^* mice than in the controls (Figure 6E,F). Therefore, the gait of the *Trkb^Penk-KO^* mice was affected, especially when pertaining to the left hind paw. 

Altogether, these data indicate that the disruption of BDNF–TrkB signalling in a specific subset of GCs derived from enkephalinergic precursors is sufficient to produce ataxia symptoms in mice.

## 4. Discussion

In this study, we investigated whether dysfunctional BDNF–TrkB signalling restricted to a specific subpopulation of cerebellar GCs is sufficient to evoke ataxia symptoms in mice. This was achieved through the conditional deletion of *Ntrk2* from enkephalinergic neurons, which, in the cerebellum, was restricted to GCs derived from enkephalinergic precursors in *Trkb^Penk-KO^* mice. Here, we show that, while *Ntrk2* deletion from around 37% of adult cerebellar GCs did not affect adult cerebellar morphology or the levels of selected synaptic markers, it led to an ataxic phenotype in the *Trkb^Penk-KO^* mice and age-dependent PC dysfunction. These findings suggest that dysfunctional BDNF–TrkB signalling in cerebellar GCs is sufficient to initiate ataxia, which has relevance to the pathophysiology of ataxia disorders characterised by disrupted BDNF–TrkB signalling, such as SCA6 and SCA1.

Reduced BDNF expression in the SCA6 cerebellum [5] and reduced BDNF protein in the SCA1 cerebellum [7] implicate BDNF–TrkB signalling in the pathophysiology of these ataxia disorders. Moreover, BDNF or BDNF mimetics have therapeutic benefits for cerebellar dysfunction in these disorders. BDNF mRNA was reduced in the cerebellum of *Atxn1^154Q/2Q^* mice (a mouse model of SCA1) at the early symptomatic stage, and the pharmacological delivery of recombinant BDNF into the lateral ventricle during the early disease stage improved the rotarod performance and ameliorated the PC pathology, as was seen from the larger molecular layer thickness and normal calbindin intensity in the treated *Atxn1^154Q/2Q^* mice [7]. Similarly, the BDNF intensity was decreased in all three cerebellar cortical layers of pre-onset SCA6^84Q/84Q^ mice (a mouse model of SCA6), and the oral administration of a TrkB agonist, 7,8-DHF, before disease onset improved the rotarod performance and elevated the PC firing frequency in these mice [10]. However, a reduction in BDNF mRNA or BDNF intensity was not shown to be restricted to the synapses on PCs in the cerebella of these mouse models; in pre-onset SCA6^84Q/84Q^ mice, BDNF intensity was shown to be reduced in the GC layer [10], and we similarly found that *NTRK2* was reduced in the GCs and not PCs of AT patients [11]. Likewise, neither BDNF delivery into the lateral ventricle nor the oral administration of 7,8-DHF restricted the elevation of TrkB signalling to PCs. Therefore, the detrimental effects of dysfunctional BDNF–TrkB signalling and the therapeutic effects of treatments that improve TrkB signalling in *Atxn1^154Q/2Q^* mice and SCA6^84Q/84Q^ mice may have been mediated by the dysfunction and restoration of this pathway in GCs, respectively. This view is strengthened by the fact that most adult GCs express *Ntrk2*, contrasting the minority of adult PCs [12]. Dysfunctional BDNF–TrkB signalling in GCs would be expected to adversely impact PC function because of the abundance of connections between the two major cell types [13] and the reliance of PC development and function on GCs [14,15,16]. Finally, dysfunctional PC would, in turn, cause impaired rotarod performances of *Atxn1^154Q/2Q^* mice and SCA6^84Q/84Q^ mice. Our findings support this view by showing that disrupted BDNF–TrkB signalling in just a subset of GCs is sufficient to initiate ataxia symptoms. Therefore, dysfunctional BDNF–TrkB signalling in GCs may contribute to the ataxia symptom in disorders such as SCA1 and SCA6. 

Previous studies have shown that GC dysfunction can lead to ataxia symptoms in mice. For example, studies disrupting PF–PC synaptic transmission to a variable extent demonstrated that the signals provided to PCs by GCs are necessary for balance and motor learning [16,42,43,44]. However, traditionally, SCAs are believed to be caused mostly by PC dysfunction, as PC degeneration is one of the most prominent pathologies in postmortem patient cerebella [2,3]. Nonetheless, in the case of SCA6, which is caused by a CAG-repeat expansion mutation in the *Cacna1a* gene, coding for the alpha 1A-voltage-dependent calcium channel [45], the *Cacna1a* gene is expressed in both PCs and GCs [46] and GC degeneration is observed in the SCA6 cerebellum [47,48]. However, little is known about the early pathological changes in GCs of SCA6 contrasted with those in PCs, with GC function not explored in SCA6 models such as SCA6^84Q/84Q^ mice [49,50,51,52] and MPI-118Q mice [53] and some mouse models purposefully restricting the SCA6 mutation to PCs [54]. Therefore, while GC dysfunction may still contribute to or even be a primary trigger of ataxia symptoms in SCA6, its role has not yet been described or studied in these mouse models.

This study enabled us to explore the role of BDNF–TrkB signalling in a subpopulation of cerebellar GCs. BDNF–TrkB signalling has been previously implicated in multiple aspects of cerebellar development and adult cerebellar function. The deletion of *Ntrk2* from all cerebellar cells (*Wnt1Cre;fBZ/fBZ)* delayed GC migration [39], led to the abnormal pruning of climbing fibre–PC synapses and decreased branching of PC dendrites [55] and interfered with the assembly and maintenance of inhibitory synapses in the molecular and GC layers [39,40]. At the same time, gross cerebellar morphology, as well as the number and differentiation of GCs, was not affected in these mutants [39,55], thus suggesting that GC survival and at least some aspects of their development do not depend on BDNF–TrkB signalling. Notably, *Ntrk2* mutant (*Wnt1Cre;fBZ/fBZ*) mice were ataxic, as seen from the paw print and accelerated rotarod tests [39,55], supporting the idea that dysfunctional BDNF–TrkB signalling can contribute to ataxia symptoms. However, whether these consequences of cerebellar *Ntrk2* deletion can be attributed specifically to BDNF–TrkB signalling ablation in GCs was not fully clear. There have been very few studies that have focused on the effects of BDNF–TrkB signalling in just GCs while limiting the influence of this signalling pathway on the other cerebellar cells. *Ntrk2* deletion from *Gabra6*-expressing cells, i.e., mature cerebellar GCs [56], did not affect the number or morphology of GCs in the mouse cerebellum [40]. While it did not influence GAD67 localisation in the cerebellar glomeruli, this GC-specific *Ntrk2* deletion resulted in reduced gephyrin localisation and a reduced number of inhibitory synapses per glomerulus in the mouse GC layer [40]. These effects were likely mediated by BDNF release from mossy fibres [57], thus suggesting that BDNF–TrkB signalling in GCs promotes the formation of inhibitory synapses between GCs and Golgi cells. In agreement with this, here, we report unaltered levels of GAD67 in the cerebella of *Trkb^Penk-KO^* mice. Interestingly, the PSD95 and synaptophysin levels were unaffected in the *Trkb^Penk-KO^* cerebella. However, as we used whole-tissue lysate, these changes, or lack thereof, cannot be localised to specific synapses. Whether the *Gabra6*-specific deletion of *Ntrk2* would lead to ataxia symptoms has not been previously explored. Our study allowed us to address this question, demonstrating that *Ntrk2* deletion from just a subset of GCs is sufficient to evoke ataxia symptoms. 

In this study, we investigated the function of GCs derived from enkephalinergic GC precursors. Not much is known about the role of enkephalin in GC precursors or the function of enkephalinergic GC precursors. It has been previously shown that Met-enkephalin is found in the EGL of P10 [18,20] and P14 [19] rats. Similarly, *Penk* mRNA was found in the EGL of rats at birth [58]. Both for Met-enkephalin and *Penk* mRNA in the EGL, the signal appeared to be the strongest in the cells near the pial surface as compared to the cells adjacent to the molecular layer [20,58], thus suggesting that preferentially transit-amplifying GC precursors, rather than post-mitotic pre-migratory GCs, are enkephalinergic, which, in turn, implies that enkephalin has a primarily developmental role in GC lineage. Our exploration of *Penk* expression in transcriptomic datasets from developing and adult mouse cerebella [12,21] confirmed that *Penk* is predominantly expressed in transit-amplifying GC precursors, as it was mainly expressed in GCs that still expressed *Atoh1*, a marker of glutamatergic progenitors, and that *Penk* was not expressed by adult GCs. Additionally, an analysis of transcriptomic datasets revealed that *Penk* was only expressed by a subset of these GC precursors. Genetic fate mapping using the BAC-*Penk-Cre* transgene corroborated these findings, as during development and in adulthood, only a subset of GCs and their precursors were descendants of the *Penk* lineage, accounting for approximately 37% of the GCs in adult cerebella. Furthermore, we demonstrated that GCs derived from these enkephalinergic precursors are involved in the cerebellar circuits responsible for balance control, although they are not necessarily the only GCs involved in these circuits. 

One limitation of our study is that, in *Trkb^Penk-KO^* mice, *Ntrk2* has also been deleted from striatal enkephalinergic neurons, an aspect that allowed for an investigation of the role of dysfunctional BDNF–TrkB signalling in Huntington’s disease (HD), a neurodegenerative disease characterised by striatal dysfunction [59], in a previous study [17]. However, it is unlikely that striatal dysfunction affected the ledge test performance of these mice, as, at 3 months, they do not experience spontaneous hyperlocomotion yet, their muscle strength is normal, and, most importantly, these mice are not impaired on the accelerating rotarod [17], a test that is traditionally used to measure striatal and cerebellar dysfunction. The gait changes in 5-month-old *Trkb^Penk-KO^* mice have been previously studied [17]. In this study, we focused on the gait parameters that are affected in patients with cerebellar ataxia [41], and expanded our cohort’s age to 3 months, the age when ledge deficit is already present. Based on this, we can conclude that the changes in gait we saw here are related to the cerebellum. At the same time, HD is a polyQ disorder [59] like SCA1 and SCA6, and it is likewise characterised by decreased BDNF–TrkB signalling [60,61]. Therefore, it is of interest that cerebellar ataxia is a common [62] and early [63] symptom of HD and cerebellar atrophy is observed in some HD cases [64], whereas, in addition to cerebellar atrophy, SCA1 and SCA6 patients present with striatal shrinkage [65]. Moreover, due to the reciprocal connections between the cerebellum and the basal ganglia [66,67,68,69], cerebellar dysfunction may lead to striatal dysfunction and vice versa. Thus, both striatal and cerebellar dysfunction may contribute to the motor impairments of patients with these polyQ disorders. Our research involving *Trkb^Penk-KO^* mice demonstrates that both the striatum and the cerebellum may be dysfunctional as a consequence of reduced BDNF–TrkB signalling in specific neuronal types in these brain areas, raising the possibility that a common pathway is responsible for some of the motor symptoms present in polyQ disorders such as HD, SCA1, and SCA6.

The findings from this study may aid in developing symptomatic treatments for ataxia (imbalance) in disorders presenting with dysfunctional cerebellar BDNF–TrkB signalling, such as SCA6 and SCA1.

## 5. Conclusions

This study shows that ataxia can be induced by the depletion of BDNF–TrkB signalling in a specific group of cerebellar GCs. As a decrease in cerebellar BDNF–TrkB signalling has been observed in spinocerebellar ataxias, including SCA1 and SCA6, our findings suggest that the dysfunction of GCs may contribute to the development of ataxia symptoms of these disorders.

## Figures and Tables

**Figure 1 biology-13-00637-f001:**
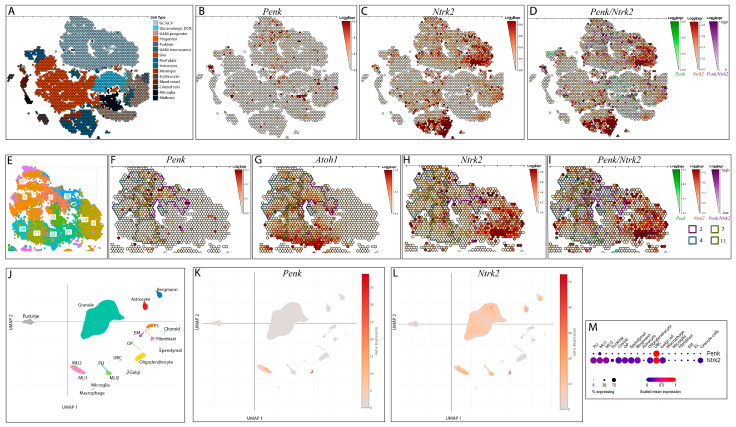
Expression of *Penk* and *Ntrk2* in developing and adult mouse cerebella through scRNA-Seq analysis. (**A**–**I**) t-SNE visualisation of the publicly available scRNA-Seq dataset [21] from developing (E10-P10) mouse cerebella (**A**–**D**), as well as zoomed-in images of the cluster of GCs and their precursors (**E**–**I**) in the t-SNE visualisation, coloured by cell identity (**A**), assigned subpopulation (**E**), and the log-normalised expression of *Penk* (**B**,**F**), *Ntrk2* (**C**,**H**), *Atoh1* (**G**), and the co-expression of *Penk* and *Ntrk2* (**D**,**I**), with selected subpopulations annotated using coloured borders (**F**–**I**). It appears that cluster 11 (sampled between E14 and E17) and clusters 2, 4, and 7 (sampled between P0 and P10) show the highest relative *Penk* expression (**F**). Cells in these clusters also express glutamatergic progenitor marker *Atoh1* (**G**), suggesting that these cells are precursors of GCs, and *Ntrk2* (**H**), suggesting that these cells are expected to be functionally affected by *Ntrk2* deletion during cerebellar development. Each hexagon represents a closely related group of cells. The expression scale is a heatmap depicting the log_2_ of the expression value. Note that the expression scale differs between individual panels. In (**D**,**I**), *Penk* expression is depicted in the shades of green and *Ntrk2* expression in the shades of red, whereas cells co-expressing the two genes are depicted in the shades of purple. The expression scale for *Penk*/*Ntrk2* co-expression is the overlay of the expression scales of the individual genes. The legend listing the colours assigned to selected subpopulations is in the panel (**I**). (**J**–**L**) UMAP visualisation of the publicly available snRNA-Seq dataset [12] from adult (P90) mouse cerebella, coloured by cell identity (**J**), and log-normalised expression of *Penk* (**K**) and *Ntrk2* (**L**). (**M**) Dot plot of scaled expression of *Penk* and *Ntrk2* in different cell types of adult (P90) mouse cerebella based on snRNA-Seq data [12]. In the adult cerebellum, GCs no longer express *Penk* (**K**,**M**), while the majority (69%) still express *Ntrk2* (**L**,**M**). Golgi cells express high levels of both *Penk* and *Ntrk2* (**K**–**M**). Scaling is relative to each gene’s expression across all cells. GC—granule cells, GCP—granule cell precursors, DCN—deep cerebellar nuclei, EM—endothelial mural, ES—endothelial stalk, OP—oligodendrocyte precursor, UBC—unipolar brush cells, PLI—Purkinje layer interneuron, and MLI1, 2—molecular layer interneuron 1, 2.

**Figure 2 biology-13-00637-f002:**
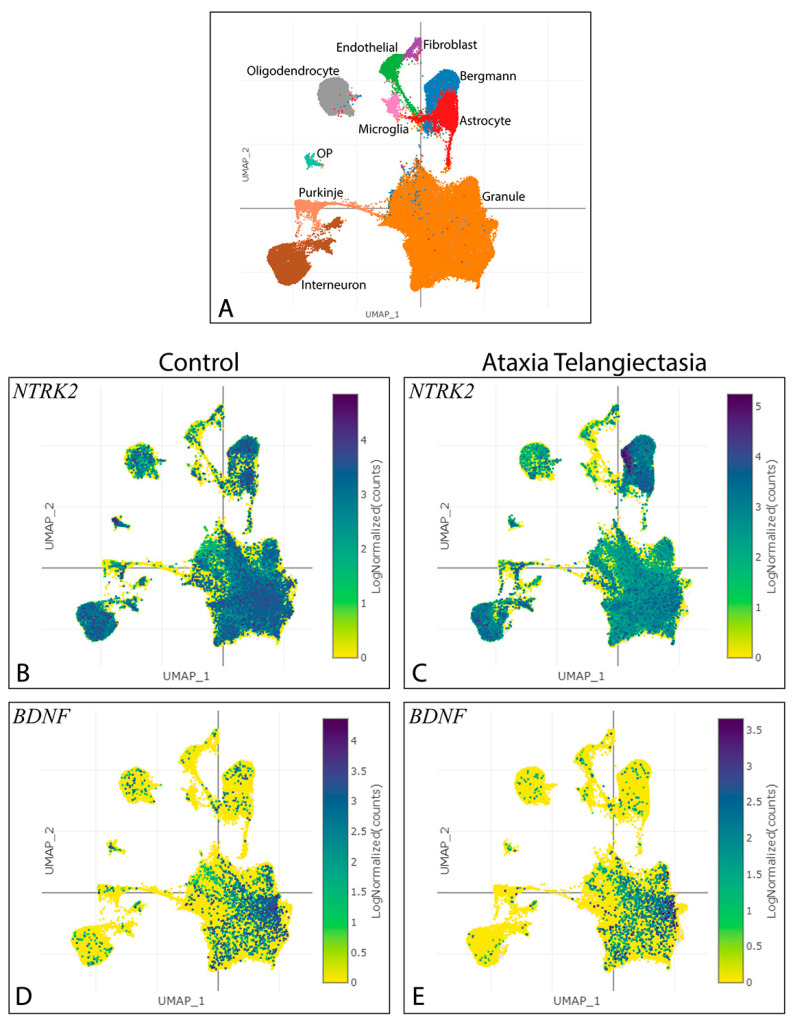
Expression of *NTRK2* and *BDNF* in cerebella of patients with ataxia–telangiectasia and healthy controls through snRNA-Seq analysis. (**A**–**E**) UMAP visualisation of the publicly available snRNA-Seq dataset [11] from cerebellar vermis of adult patients with AT and healthy controls, coloured by cell identity (**A**), and log-normalised expression of *NTRK2* (**B**,**C**) and *BDNF* (**D**,**E**). *NTRK2* expression appears to be decreased specifically in the GCs of AT patients compared to those of healthy controls (**B**,**C**). *BDNF* expression does not appear to be particularly affected in any of the cell types of AT patients (**D**,**E**). Scaling is relative to each gene’s expression across all cells. OP—oligodendrocyte precursor.

**Figure 3 biology-13-00637-f003:**
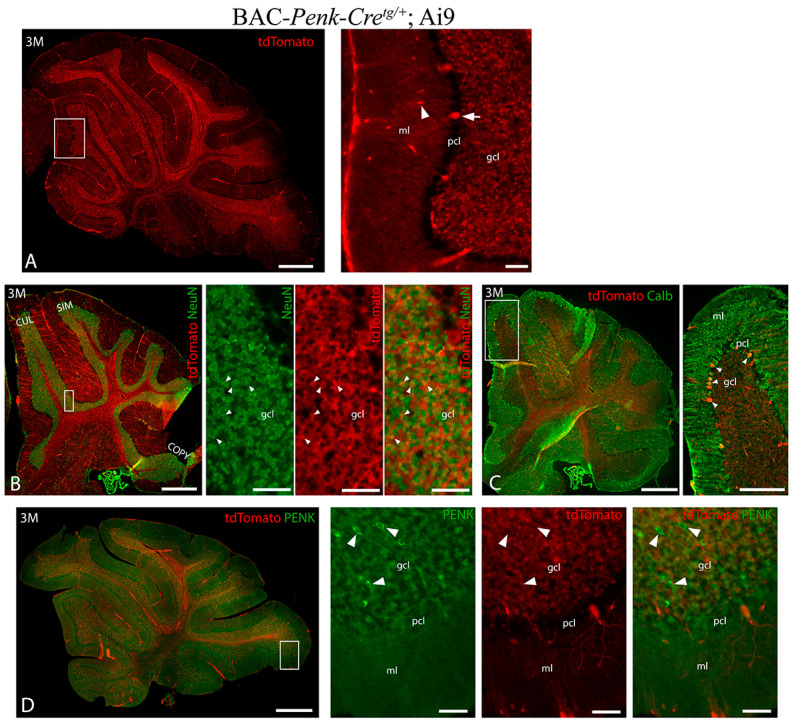
NeuN immunostaining revealed that only a subset of GCs expressed tdTomato in the adult cerebella of BAC-*Penk*-*Cre* mice. (**A**) Three-month-old representative sagittal cerebellar section of a BAC-*Penk*-*Cre^tg^*^/+^;Ai9 mouse reveals that Cre-mediated tdTomato expression occurred primarily in the GC layer. The arrowhead and the arrow indicate tdTomato+ molecular layer interneuron and Purkinje cell, respectively. (**B**) NeuN immunostaining (green) on a sagittal cerebellar section of a 3 M BAC-*Penk*-*Cre^tg^*^/+^;Ai9 mouse demonstrates that only a subset of GCs (yellow) undergoes Cre-recombination. Arrowheads indicate tdTomato+ GCs. (**C**) Calbindin immunostaining (green) on a representative sagittal cerebellar section of a 3 M BAC-*Penk*-*Cre^tg^*^/+^;Ai9 mouse demonstrates that only a few PCs underwent Cre-recombination. Arrowheads indicate tdTomato+ PCs. (**D**) PENK immunostaining (green) on a sagittal cerebellar section of a 3 M BAC-*Penk*-*Cre^tg^*^/+^;Ai9 mouse shows no PENK in the cerebellar GCs at this stage. Arrowheads indicate enkephalinergic tdTomato− Golgi cells. Scale bars: 500 μm in (**A**,**B**,**D**) and 50 μm in respective insets; 500 μm in (**C**) and 200 μm in the respective inset. gcl—granule cell layer, pcl—Purkinje cell layer, ml—molecular layer; CUL—culmen, SIM—simple lobule, and COPY—copula pyramidis.

**Figure 4 biology-13-00637-f004:**
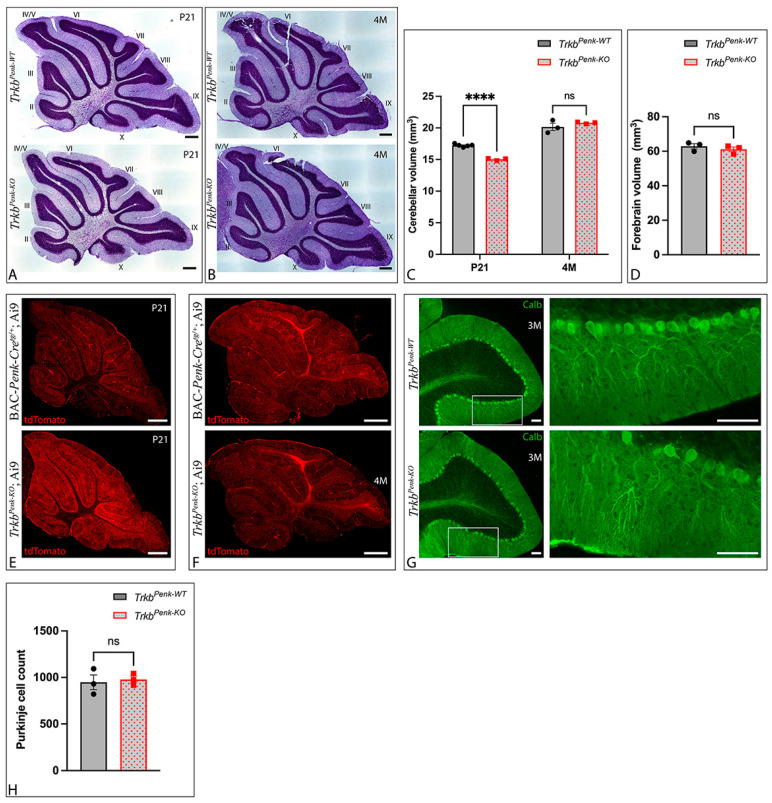
Cerebellar development is delayed in *Trkb^Penk-KO^* mice, (**A**,**B**) Representative images of cresyl-violet-stained cerebellar sections from P21 (**A**) and 4 M (**B**) control and mutant mice demonstrate no apparent gross abnormalities in the cerebellar lobular structure. The lobule numbers are indicated by Roman numerals. (**C**) Cavalieri analysis of P21 and 4 M cerebella revealed cerebellar development was delayed in *Trkb^Penk-KO^* mice. P21 mutant cerebella were significantly smaller than controls’, *t*(6) = 13.71, adjusted *p* < 0.0001. Controls, *n* = 5, 3 males and 2 females; mutants, *n* = 3, 2 males and 1 female. At 4 M, the difference was no longer significant, *t*(4) = 1.110, adjusted *p* = 0.658. Controls, *n* = 3; mutants, *n* = 3. (**D**) 1 M forebrain volumes did not differ between control and mutant mice, suggesting that delayed cerebellar development of *Trkb^Penk-KO^* mice (**C**) did not generalise to other brain areas, *t*(4) = 0.841, *p* = 0.448. Controls, *n* = 3, 2 males and 1 female; mutants, *n* = 3, all males. (**E**,**F**) Representative images of cerebellar sections from P21 (**E**) and 4 M (**F**) BAC-*Penk-Cre^tg/+^*;Ai9 and *Trkb^Penk-KO^*;Ai9 mice reveal that recombined GCs are intact in the mutant mice. Scale bars: 500 μm. (**G**) Calbindin-stained sections from 3 M *Trkb^Penk-WT^* and *Trkb^Penk-KO^* mice show that the dendritic trees of PCs are largely unaffected in mutant mice. Scale bars: 75 μm. (**H**) Total PC count of 4 M *Trkb^Penk-KO^* mice and controls did not differ, *t*(4) = 0.345, *p* = 0.748. Controls, *n* = 3, all females; mutants, *n* = 3, all females. **** *p* ≤ 0.0001; ns, not significant (*p* > 0.05).

**Figure 5 biology-13-00637-f005:**
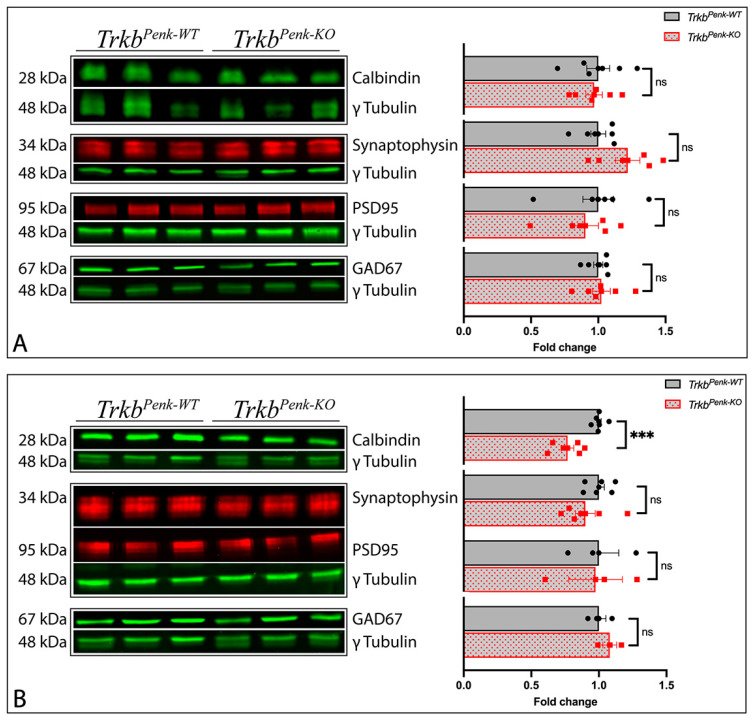
Molecular changes in *Trkb^Penk-KO^* cerebella. (**A**,**B**) Western blot analyses of calbindin and synaptic proteins in cerebellar lysates from 3 M (**A**) and 8 M (**B**) *Trkb^Penk-WT^* and *Trkb^Penk-KO^* mice. (**A**) At 3 M, calbindin levels were not affected in the cerebellum of *Trkb^Penk-KO^* mice, *t*(10) = 0.291, *p* = 0.777. Similarly, the synaptophysin levels in *Trkb^Penk-KO^* cerebella were not significantly different from those of controls, *t*(10) = 2.072, *p* = 0.065. The levels of other tested synaptic proteins were also not affected (PSD95, *t*(10) = 0.642, *p* = 0.535; GAD67, *t*(10) = 0.302, *p* = 0.769). Controls, *n* = 6, 3 males and 3 females; mutants, *n* = 6, 3 males and 3 females. (**B**) At 8 M, calbindin levels were significantly decreased in the cerebellum of *Trkb^Penk-KO^* mice, *t*(10) = 4.664, *p* < 0.001. There was no difference in the levels of the synaptic marker synaptophysin in the cerebellum of *Trkb^Penk-KO^* mice, *t*(10) = 1.197, *p* = 0.259, controls, *n* = 6, 1 male and 5 females; mutants, *n* = 6, 1 male and 5 females. The levels of PSD95 and GAD67 remained unchanged (PSD95, *t*(4) = 0.101, *p* = 0.925; GAD67, *t*(4) = 1.109, *p* = 0.330). Controls, *n* = 3, 1 male and 2 females; mutants, *n* = 3, 1 male and 2 females. *** *p* ≤ 0.001; ns, not significant (*p* > 0.05). Full Western blot figures can be viewed in Appendix A.

**Figure 6 biology-13-00637-f006:**
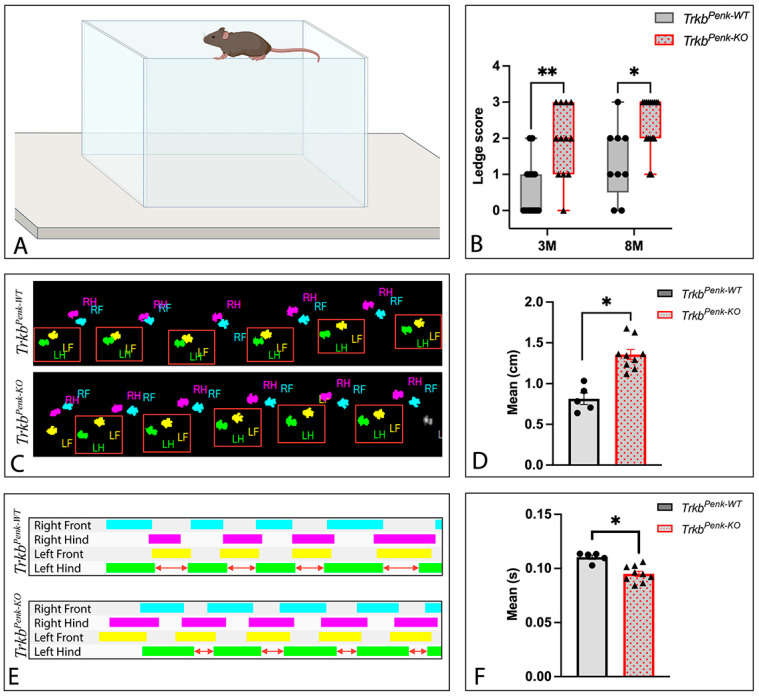
Motor coordination and gait were altered in *Trkb^Penk-KO^* mice at the early stage. (**A**) Illustration of a mouse performing the ledge test. (**B**) Ledge scores of 3 M and 8 M *Trkb^Penk-KO^* mice and controls, where 3 M mutants received significantly higher (worse) ledge scores than controls (Mann–Whitney *U* = 25.50, adjusted *p* = 0.005), suggesting a balance impairment. Controls, *n* = 13, 6 males and 7 females; mutants, *n* = 12, 7 males and 5 females. The difference persisted at 8 M (Mann–Whitney *U* = 28.00, adjusted *p* = 0.014). Controls, *n* = 9, 2 males and 7 females; mutants, *n* = 16, 6 males and 10 females. Different cohorts of animals were tested at 3 M and 8 M. (**C**) CatWalk footprints of a 3 M *Trkb^Penk-KO^* mouse and a 3 M control. (**D**) Print positions of left paws (distance between paws framed in red in (**C**)) were significantly elevated in 3–5 M *Trkb^Penk-KO^* mice, *t*(9.68) = 5.721, adjusted *p* = 0.015. Controls, *n* = 5, 2 males and 3 females; mutants, *n* = 9, 6 males and 3 females. (**D**) CatWalk gait diagrams of a 3 M *Trkb^Penk-KO^* mouse and a 3 M control. (**F**) Left hind swing (duration of no contact of the left hindpaw with the glass plate, indicated by red arrows in (**E**)) was significantly reduced in 3–5 M *Trkb^Penk-KO^* mice, *t*(11.74) = 4.823, adjusted *p* = 0.029. Controls, *n* = 5, 2 males and 3 females; mutants, *n* = 9, 6 males and 3 females. Individual paws are indicated by unique colours: right front, blue; right hind, pink; left front, yellow; left hind, green. LH—left hindpaw, LF—left frontpaw, RH—right hindpaw, and RF—right frontpaw. ** *p* ≤ 0.01, * *p* ≤ 0.05. Figure 6A was created with BioRender.com (accessed on 1 July 2024).

**Table 1 biology-13-00637-t001:** Expression of *Penk* and *Ntrk2* in the GC lineage of developing mouse cerebella. Data extracted from the re-analysed [31] scRNA-Seq dataset [21] of E10–E17 mouse cerebella.

	*Atoh1*	*Pax6*	*Penk*+ Glutamatergic Progenitors, GC Precursors and GCs
Total cells	10,706	12,538	169
*Penk*	187	162	N/A
*Penk* % of cells	1.75	1.29	N/A
*Ntrk2*	1352	2686	25
*Ntrk2* % of cells	12.63	21.42	14.79

**Table 2 biology-13-00637-t002:** Expression of *Penk* and *Ntrk2* in selected cell types of P90 mouse cerebella. Data extracted from the snRNA-Seq dataset [12] of adult (P90) mouse cerebella. GCs—granule cells and PCs—Purkinje cells.

	GCs	Golgi Cells	PCs
Total cells	477,176	3989	16,634
*Penk* scaled expression	0.01	3.85	0.02
*Penk* % of cells	1.03	81.35	2.03
*Ntrk2* scaled expression	1.48	11.17	0.4
*Ntrk2* % of cells	69.47	99	26.72

**Table 3 biology-13-00637-t003:** Hindpaw clasping and kyphosis test scores of 3 M and 8 M *Trkb^Penk-KO^* mice and controls. Hindpaw clasping and kyphosis scores were normal in 3 M and 8 M *Trkb^Penk-KO^* mice.

Test	Genotype	Age (in Months)	*n*	Median	Max	Min	Mann-Whitney *U*	Adjusted *p*
Hindpaw clasping	Control	3	9, all male	0	0	0	40.50	>0.999
	Mutant	3	9, all male	0	0	0		
	Control	8	3, 1 male and 2 females	0	0	0	6.00	>0.999
	Mutant	8	4, 2 males and 2 females	0	0	0		
Kyphosis test	Control	3	10, 3 males and 7 females	0	0	0	45.00	>0.999
	Mutant	3	9, 4 males and 5 females	0	0	0		
	Control	8	6, 1 male and 5 females	0	0	0	33.00	>0.999
	Mutant	8	12, 4 males and 8 females	0	2	0		

## Data Availability

Data are available upon request.

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
