# Peer review of "Ablation of TrkB from Enkephalinergic Precursor-Derived Cerebellar Granule Cells Generates Ataxia"

_biology, 2024, doi:10.3390/biology13080637_

Round 1

Reviewer 1 Report

Comments and Suggestions for Authors

The present manuscript entitled "Ablation of TrkB from enkephalinergic precursor-derived cerebellar granule cells generates ataxia. " by Elena Eliseeva, Mohd Yaseen Malik and Dr Liliana Minichiello is well written with clear figures. The results while preliminary suggest that a dysfunction of the BDNF-TrkB signaling in cerebellar granule neurons might be involved in the initiation of ataxia in mice. The genetic and phenotypic work are convincing. I however have comments and suggestion regarding the results and their interpretation: 

It is surprising that the authors do not perfuse the animal with PBS before the perfusion of PFA. Is this an omission in the method section? 

Please be consistent in the naming of the reagents: e.g. TritonX Triton X100 

Please provide a reference for ImageJ 

The information regarding the anesthesia procedure is missing 

The setting for the sonication of the samples is missing 

What kind of gel was used for electrophoresis? 

Does Penk expression level change in the cerebellum of ataxic mice? 

NeuN is not specific for mature GC it is a marker for many neuronal types. Neurod1 might be more specific for GN but other markers have been described.  

Did the cerebellum development delay apply also to other parts of the brain? Were the mice smaller? 

The author should consider a less definitive conclusion regarding cerebellar synaptic impairment as they only look at one marker using western blot and did not perform other type of experiment to validate this observation. Such validation would improve the quality of the manuscript. 

Did the PC density was affected? Could change be observed only in certain lobules? The Progressive loss of PC can be lobule specific? If a difference exists, can it be linked to the pattern observed in the Tomato model? 

The pathology at 8months is missing. Why only include the phenotypic results and not the matching histology? 

The defect suggested in the present study may expand the HD but what about other ataxia. scRNAseq from other mouse models are available and could be looked at to see if the findings generalize. If so, this would strengthen the findings.

Author Response

Response to Reviewer 1 Comments

1. Summary

We would like to thank the reviewers for taking the time to provide constructive feedback and insightful remarks, which helped us improve our manuscript. We have revised the manuscript by incorporating new experiments and analyses, which are described in responses to the reviewers. Revisions are highlighted in the re-submitted manuscript and are marked in red in the responses.

2. Point-by-point response to Comments and Suggestions for Authors

Comments 1: The present manuscript entitled "Ablation of TrkB from enkephalinergic precursor-derived cerebellar granule cells generates ataxia. " by Elena Eliseeva, Mohd Yaseen Malik and Dr Liliana Minichiello is well written with clear figures. The results while preliminary suggest that a dysfunction of the BDNF-TrkB signaling in cerebellar granule neurons might be involved in the initiation of ataxia in mice. The genetic and phenotypic work are convincing. I however have comments and suggestion regarding the results and their interpretation: It is surprising that the authors do not perfuse the animal with PBS before the perfusion of PFA. Is this an omission in the method section? 

Response 1: The reviewer is correct; we omitted this information. We have now included it in the 2.2 Tissue processing subsection of the Methods [page 4, line 111].

Comments 2: Please be consistent in the naming of the reagents: e.g. TritonX Triton X100 

Response 2: We thank the reviewer for pointing this out. We have ensured that TritonX-100 and Tween-20 are consistently named in the 2.3 Immunofluorescence and 2.7 Immunoblotting subsections of the Methods [page 4, lines 123, 127; page 6, lines 215, 217].

Comments 3: Please provide a reference for ImageJ 

Response 3: As suggested by the reviewer, we have added the reference for ImageJ in the 2.4 Estimation of a proportion of GCs and PCs undergoing recombination and 2.6 Nissl staining and Cavalieri analysis subsections of the Methods [page 5, lines 138, 178].

Comments 4: The information regarding the anesthesia procedure is missing 

Response 4: We are grateful to the reviewer for this suggestion. We have included the details of the anaesthesia procedure to the manuscript. The revised text reads as follows in the 2.2 Tissue processing subsection of the Methods: “After intraperitoneal injection with pentobarbital (50mg/kg)” [page 4, line 110], and in the 2.7 Immunoblotting subsection of the Methods: “Mice were sacrificed by cervical dislocation after anaesthesia (intraperitoneal injection of pentobarbital, 50mg/kg)” [page 6, line 205].

Comments 5: The setting for the sonication of the samples is missing 

Response 5: We thank the reviewer for pointing this out. We have modified the text in the 2.7 Immunoblotting subsection of the Methods: “After 5-minute sonication (30s on/30s off) and centrifugation” [page 6, line 209].

Comments 6: What kind of gel was used for electrophoresis? 

Response 6: In the 2.7 Immunoblotting subsection of the Methods, we state that “Total protein lysates were then resolved on SDS-PAGE gels” [page 6, line 213] and that “the corresponding gel percentages <…> are listed in Supplementary Table 2” [page 6, line 218].

Comments 7: Does Penk expression level change in the cerebellum of ataxic mice?

Response 7: We are grateful to the reviewer for raising this question. Our previous research has revealed that striatal enkephalin expression is reduced in TrkbPenk-KO mice (Ref.17, Besusso et al., Nat Commun 2013). Differently from striatal enkephalinergic neurons, there is very little PENK in the cerebellum of adult mice (see below excerpts from originally Figure 1, now Figure 2, of the manuscript), which is why we did not compare cerebellar PENK levels in TrkbPenk-KO mice and littermate controls.

Figure 2. (J-L) UMAP visualisation of the publicly available snRNA-Seq dataset [12] from adult (P90) mouse cerebella, coloured by cell identity (J), and log-normalised expression of Penk (K) and Ntrk2 (L). (M) Dot plot of scaled expression of Penk and Ntrk2 in different cell types of adult (P90) mouse cerebellum based on snRNA-Seq data [12]. In the adult cerebellum, GCs no longer express Penk (K, M), while the majority (69%) still express Ntrk2 (L, M). Golgi cells express high levels of both Penk and Ntrk2 (K-M). Scaling is relative to each gene's expression across all cells. GC – granule cells, GCP – granule cell precursors, DCN – deep cerebellar nuclei, EM – endothelial mural, ES – endothelial stalk, OP – oligodendrocyte precursor, UBC –unipolar brush cells, PLI – Purkinje layer interneuron, MLI1, 2 – molecular layer interneuron 1, 2.

Legend: Expression of PENK in the cerebella of patients with AT and healthy controls through snRNA-Seq analysis. (A-C) UMAP visualisation of the publicly available snRNA-Seq dataset (Ref.11, Lai, J. et al., (2024). Cell reports, 43(1)) from cerebellar vermis of adult patients with ataxia-telangiectasia (C) and healthy controls (B), coloured by cell identity (A), and log-normalised expression of PENK (B, C). It appears that granule cells of ataxia-telangiectasia patients express more PENK than those of healthy controls.

However, by exploring a publicly available transcriptomic dataset from patients with ataxia telangiectasia (AT) and healthy controls (Ref.11,Lai, J. et al., (2024). Cell reports43(1)), we can observe that PENK levels may actually be elevated in the granule cells of AT patients. Image provided below.

Although this is an interesting subject, it requires further investigation, and it would be out of the scope of this present manuscript.

Comments 8: NeuN is not specific for mature GC it is a marker for many neuronal types. Neurod1 might be more specific for GN but other markers have been described.  

Response 8: We agree that NeuN is a marker of many different neurons, however, in the cerebellum, is restricted to the mature granule cells as reported in the literature Ref.33 (Weyer, A.; Schilling, K. Developmental and cell type-specific expression of the neuronal marker NeuN in the murine cerebellum. Journal of Neuroscience Research 2003, 73, 400–409, doi:10.1002/jnr.10655).

Comments 9: Did the cerebellum development delay apply also to other parts of the brain? Were the mice smaller? 

Response 9: This is an important point. To address it, we have compared forebrain volumes of one-month-old mice and found that there was no difference in forebrain volume at this stage. This was included in subsection 3.3 Unaltered cerebellar morphology of TrkbPenk-KO mice of the Results [page 15, lines 515-517] and Figure 4(D) [page 16] of the manuscript (see below).

Figure 4. (D) 1M forebrain volumes did not differ between control and mutant mice, suggesting that delayed cerebellar development of TrkbPenk-KO mice (C) did not generalise to other brain areas, t(4)=0.841, p=0.448. Controls, n=3; mutants, n=3.

Additionally, we compared the weight of mice at P21, P29, 3 months and 8 months. There was no difference in weight at any stage, suggesting that the delay in development was not generalised. These results were included in the 3.3 Unaltered cerebellar morphology of TrkbPenk-KO mice subsection of the Results [page 15, lines 514-515] and Supplementary Figure 6.

Supplementary Figure 6. Body weight analysis across different age groups. Comparison analysis revealed no differences in body weight of TrkbPenk-KO mice and controls across various age groups. Data from both sexes were pooled together for analysis as there was no interaction between sex and genotype for any age group analysed, (P21, n = 9 controls (3 males and 6 females), and 4 mutants (3 males and 1 female), t(11)=0.745, p=0.472; P29, n = 10 controls (5 males and 5 females), and 5 mutants (3 males and 2 females), t(13)=0.106, p=0.917; 3M, n = 20 controls (9 males and 11 females), and 17 mutants (9 males and 8 females), t(35)=0.678,  p=0.503; 8M, n = 13 controls (6 males and 7 females), and 18 mutants (8 males and 10 females), t(29)=0.126, p=0.901. Values are means ± SEM. p statistic from unpaired, two-tailed, Student’s t-test.

Comments 10: The author should consider a less definitive conclusion regarding cerebellar synaptic impairment as they only look at one marker using western blot and did not perform other type of experiment to validate this observation. Such validation would improve the quality of the manuscript. 

Response 10: We appreciate the reviewer's suggestion. To address this, we conducted additional experiments. First, we added additional cohorts and performed a time-course analysis, including the 3- and 8-month stages. Secondly, we added an additional synaptic marker, PSD95, a postsynaptic density marker, to broaden our analysis of potential synaptic defects. In addition, we purchased a different synaptophysin antibody, which is highly regarded for its use in western blotting (Anti-Synaptophysin antibody [ab52636]). It is a rabbit recombinant monoclonal antibody, offering several advantages, including high batch-to-batch consistency and reproducibility. It also offers improved sensitivity and specificity. Therefore, our final results on this matter have indicated that the differences in synaptophysin levels were not significantly different between mutants and controls at 3 or 8 months using different cohorts [page 18, lines 570-572, 573-574]. Similarly, we extended the GAD67 analysis to the 8-month stage and explored levels of PSD95 at 3- and 8M. As shown in Fig.5 (A-B), these two markers were unaffected at these two stages. Having expanded this analysis on additional marker, stages and cohorts, we can now better conclude the following: “Therefore, based on these results, there was no evidence to suggest specific cerebellar synaptic molecular changes in TrkbPenk-KO mice at 3M and 8M.” [page 18, lines 575-576].

Comments 11: Did the PC density was affected? Could change be observed only in certain lobules? The Progressive loss of PC can be lobule specific? If a difference exists, can it be linked to the pattern observed in the Tomato model? 

Response 11: These are important questions raised by the reviewers. In order to answer them, we performed Purkinje cell counting in 4-month-old mice (3 controls, 3 mutants). We found that the total number of Purkinje cells, estimated as the sum of Purkinje cells in the simple lobule, lobules II and III, culmen and nodulus, was not significantly different between controls and mutants. Additionally, individual Purkinje cell counts specifically in lobule III, culmen and nodulus support the conclusion. This was included in subsection 3.3 Unaltered cerebellar morphology of TrkbPenk-KO mice of the Results [page 17, lines 538-543], Figure 4(H) [page 16] and Supplementary Figure 7 of the manuscript (see below).

Figure 4. (H) Total Purkinje cell count of 4M TrkbPenk-KO mice and controls did not differ, t(4)=0.345, p=0.748. Controls, n=3; mutants, n=3.

Supplementary Figure 7. Purkinje cell counts in selected lobules of TrkbPenk-KO cerebella. Scatter bar plot showing no significant differences in Purkinje cell counts of selected individual lobules at 4M between TrkbPenk-KO mice and controls. Lobule III, t(4)=1.220, p=0.290. Culmen: t(4)=0.546, p=0.614. Nodulus: t(4)=1.235, p=0.284. Controls, n=3, all females; mutants, n=3, all females. Values are means ± SEM. p statistic from unpaired, two-tailed, Student’s t-test.

Comments 12: The pathology at 8months is missing. Why only include the phenotypic results and not the matching histology? 

Response 12: We thank the reviewer for these comments. While the ageing of TrkbPenk-KO mice was not the focus of our manuscript, as our main interest is to understand the early impact derived by Ntrk2 deletion in a subpopulation of cerebellar granule neurons at the molecular and phenotypic levels, we chose to include the behavioural data from 8 months as it suggests that the deficit seen on the ledge test at 3 months would occur in controls naturally with ageing – in other words, deletion of Ntrk2 from a subset of cerebellar granule cells can lead to premature ageing-like effects on the mouse behaviour. However, in view of the reviewer’s comment, we decided to explore the molecular changes occurring in the cerebella of 8-month-old TrkbPenk-KO mice, as this would allow us to understand which elements of the cerebellar circuitry may be getting progressively dysfunctional. For example, calbindin levels are not significantly different between controls and mutants at 3 months, however, at 8 months calbindin levels are significantly decreased in mutant cerebellum, suggesting Purkinje cell dysfunction. This means that Purkinje cells are possibly impacted early on, but this more subtle dysfunction couldn’t be detected through Western blotting or cell counting.

We have not conducted electrophysiological measurements on cerebellar brain slices in the current model because we are affecting a specific subpopulation of granular neurons. This would require a more sophisticated experiment to determine the impact of this specific deletion. For instance, conducting extracellular recordings in the cerebellar cortex of alert mice would be the ideal way to conclusively test the in vivo functional outcome of this deletion and its impact on animal physiology. Although we plan to conduct such experiments in future investigations, it is not feasible for the scope of this manuscript and the time required to perform such experiments.

Therefore, extending the analysis of molecular markers such as calbindin levels at 8 months adds to our understanding of cerebellar dysfunction of TrkbPenk-KO, revealing the progressive impact of the deletion. The latter is also supported in this manuscript by the behavioural phenotypes showing impairment already at 3 months in mutants.

The results of Western blot analyses from 8-month-old mice were included in Figure 5(B) [page 17], in the 3.4 Molecular consequences of Ntrk2 deletion in TrkbPenk-KO mice subsection of the Results [page 18, lines 562-565]: “To explore whether PCs are affected with age, we compared calbindin levels in 8M mice. Interestingly, we found that calbindin levels significantly decreased by 23% in 8M TrkbPenk-KO mice (Figure 5B). Therefore, Purkinje cells may be either lost or functionally affected in 8M TrkbPenk-KO mice”.

And on page 18, lines 573-576: “<…>, we observed that synaptophysin, GAD67, and PSD95 levels remained unchanged at a later stage, 8M TrkbPenk-KO cerebella (Figure 5B). Therefore, based on these results, there was no evidence to suggest specific cerebellar synaptic molecular changes in TrkbPenk-KO mice at 3M and 8M”.

Comments 13: The defect suggested in the present study may expand the HD but what about other ataxia. scRNAseq from other mouse models are available and could be looked at to see if the findings generalize. If so, this would strengthen the findings.

Response 13: We appreciate the advice provided by the reviewer. This is a very good suggestion to generalise our findings. Accordingly, we have explored a publicly available transcriptomic dataset from the cerebella of patients with ataxia telangiectasia (AT) and healthy controls (Ref.11,Lai, J. et al., (2024). Cell reports43(1)). NTRK2 was reduced in the granule cells of AT patients, but not in the Purkinje cells. BDNF expression did not appear to be affected in the granule cells of AT patients, suggesting that TrkB signalling would be affected within the granule cells of AT patients, whereas BDNF production in the granule cells and consequently TrkB signalling in the cells receiving input from granule cells would be expected to remain unaffected. This information was included in subsection 3.1 Exploring the impact of Ntrk2 deletion in a specific subset of cerebellar granule cells

Results [page 10, lines 317-326] and Figure 1 from the revised manuscript (see also below, although a better image resolution is provided in Fig.1).

Figure 1. Expression of NTRK2 and BDNF in cerebella of patients with ataxia-telangiectasia and healthy controls through snRNA-Seq analysis. (A-E) UMAP visualisation of the publicly available snRNA-Seq dataset [11] from cerebellar vermis of adult patients with ataxia-telangiectasia (AT) and healthy controls, coloured by cell identity (A), and log-normalised expression of NTRK2 (B, C) and BDNF (D, E). NTRK2 expression appears to be decreased specifically in the GCs of AT patients compared to those of healthy controls (B, C). BDNF expression does not appear to be particularly affected in any of the cell types of AT patients (D, E). Scaling is relative to each gene's expression across all cells. OP – oligodendrocyte precursor

5. Additional clarifications

In addition to the revisions described in the responses to the reviewer, we have condensed subsections 3.2-3.4 of the Results section to improve the readability of the revised manuscript. Therefore, Figures 2 and 4 of the original manuscript were moved into Supplementary Files. Similarly, the assessment of the dendritic trees of Purkinje cells was moved to subsection 3.3, Unaltered cerebellar morphology of TrkbPenk-KO mice of the Results, as it became more appropriate after the addition of Purkinje cell counting to the subsection. Additionally, we converted the graphs presenting hindpaw clasping and kyphosis scores into one table (Table 3) to reduce the size of the revised Figure 6, which now includes the CatWalk results.

Reviewer 2 Report

Comments and Suggestions for Authors

This manuscript is the continuation of a previous study by the authors which used the same mouse model and studied the striatum. The mouse model is a conditional knockout of the TrkB receptor induced by Cre expression under the pro-enkephalin promoter. This Cre line shows expression mostly in the striatum, but also in a subset of cerebellar neurons, mostly granule cells. Based on reporter expression in a reporter line about 37% of the cerebellar granule cells expressed the reporter. This would mean that about one third of the TrkB positive granule cells would lose TrkB expression. This goes together with the Western Blot analysis demonstrating a reduction of TrkB expression by 24%. Taken together, in this mouse line there is a loss of TrkB expression in an unknown subpopulation of granule cells. Surprisingly, the authors find a 48% reduction of synaptophysin and a more than 50% reduction in Calbindin expression in the cerebellum by Western blot analysis. In contrast, immunohistochemistry for calbindin looked normal with no evidence for a reduction of expression. The mice were then tested in an ataxia scoring system including hindlimb clasping, kyphosis and a ledge test. No abnormalities were found in hindlimb clasping and kyphosis, but a reduced performance in the ledge test was found for 3 months old TrkB-Enk deficient mice.

The manuscript uses modern techniques like scRNASeq analysis and a sophisticated mouse line. Unfortunately, the findings are not really consistent and difficult to interpret. On the one hand the authors show that only a subset of granule cells is affected, i.e. about one third of the 70% granule cells that normally express TrkB would be deficient for the TrkB receptor in the mouse line. On the other hand they report rather strong effects for protein expression of Synaptophysin and Calbindin which could only be explained by a major disruption of cerebellar synapses. On the other hand, no abnormalities are found in the immunohistochemical analysis. A crucial point is the behavioral analysis which is relying on the so-called ledge test. This test is not commonly used in cerebellar research and the reported results are somewhat strange. Control mice are supposed to have a ledge score of 0, in the reported data they have a ledge score of about 1.5, the mutants of 2.5. From the previous publication it is already known that the mutants haveno deficit in the rotarod test.

Taken together, the data appear unclear and also seem to be based on a very restricted sampling. Western blot analysis was done in 3 animals each, and the ledge test with 6 animals from each genotype. With these small group sizes, the chance of unreliable results is high.

Based on presented data it is not justified to claim an ataxic phenotype. In addition, the presented cerebellar changes in the mutant mice could not explain the presence of such a phenotype. As the TrkB deletion is brain-wide in proenkephalin positive neurons it is not possible to assign the observed behavioral effects to the rather moderate cerebellar changes but it is more likely that they are due to effects in the other parts of the brain.

Therefore, the authors are advised to do the following additional experiments:

1) Repeat the Western blot analysis with a new set of mice. Finally, there should be 3 independent experiments with mice from 3 different litters.

2) Do a more extensive behavioral testing program. Rotarod is fine and has been done, add beam walking test and/or grid walking for better assessing the ataxic phenotype. The data presented so far are not convincing. Use a larger group of animals (approx. 20) for each genotype.

3) Better assess Purkinje cell survival and morphology. Try to do a quantification of Purkinje cell number in some lobules.

4) Explain the very high tdTomato expression in the meninges (Figs. 2 and 3).

5) The quality of the histological images in the pdf was very poor and no download of better-quality images was offered. This is most likely to too strong reduction during pdf production. This needs to be improved.

In addition, the manuscript would greatly benefit from doing the behavioral experiments in parallel with the mouse line with a specific deletion of TrkB in all cerebellar granule cells (Chen et al. 2011). These mice should be available to the authors as the senior author was a co-author of this publication. This would help to better assess the effects of the deletion in the pro-enkephalin positive granule cells vs. all granule cells.

Author Response

Response to Reviewer 2 Comments

1. Summary

We would like to thank the reviewers for taking the time to provide constructive feedback and insightful remarks, which helped us improve our manuscript. We have revised the manuscript by incorporating new experiments and analyses, which are described in responses to the reviewers. Revisions are highlighted in the re-submitted manuscript and are marked in red in the responses.

2. Point-by-point response to Comments and Suggestions for Authors

Comments 1: This manuscript is the continuation of a previous study by the authors which used the same mouse model and studied the striatum. The mouse model is a conditional knockout of the TrkB receptor induced by Cre expression under the pro-enkephalin promoter. This Cre line shows expression mostly in the striatum, but also in a subset of cerebellar neurons, mostly granule cells. Based on reporter expression in a reporter line about 37% of the cerebellar granule cells expressed the reporter. This would mean that about one third of the TrkB positive granule cells would lose TrkB expression. This goes together with the Western Blot analysis demonstrating a reduction of TrkB expression by 24%. Taken together, in this mouse line there is a loss of TrkB expression in an unknown subpopulation of granule cells. Surprisingly, the authors find a 48% reduction of synaptophysin and a more than 50% reduction in Calbindin expression in the cerebellum by Western blot analysis. In contrast, immunohistochemistry for calbindin looked normal with no evidence for a reduction of expression. The mice were then tested in an ataxia scoring system including hindlimb clasping, kyphosis and a ledge test. No abnormalities were found in hindlimb clasping and kyphosis, but a reduced performance in the ledge test was found for 3 months old TrkB-Enk deficient mice.

The manuscript uses modern techniques like scRNASeq analysis and a sophisticated mouse line. Unfortunately, the findings are not really consistent and difficult to interpret. On the one hand the authors show that only a subset of granule cells is affected, i.e. about one third of the 70% granule cells that normally express TrkB would be deficient for the TrkB receptor in the mouse line. On the other hand they report rather strong effects for protein expression of Synaptophysin and Calbindin which could only be explained by a major disruption of cerebellar synapses. On the other hand, no abnormalities are found in the immunohistochemical analysis.

Response 1: We are grateful for the reviewer’s insightful remarks.

Regarding the differences we previously reported for Synaptophysin and Calbindin, we have come to realize that the cohort used in Fig.6A and 6C of the previous version of this manuscript showed rather huge variability in the control group which could have been the reason for such differences. Therefore, as explained in the response to comment#5 from this reviewer, we have now addressed this point by conducting more experiments and a time course analysis at 3 and 8 months (please refer to point 5 response below).

Comments 2: A crucial point is the behavioral analysis which is relying on the so-called ledge test. This test is not commonly used in cerebellar research and the reported results are somewhat strange. Control mice are supposed to have a ledge score of 0, in the reported data they have a ledge score of about 1.5, the mutants of 2.5.

Response 2: We respectfully disagree with this view. The ledge test directly measures coordination, which is impaired in cerebellar ataxias and other neurodegenerative disorders, making it comparable to human signs of cerebellar ataxia (for example, Ref.28: Guyenet, et al., 2010. A Simple Composite Phenotype Scoring System for Evaluating Mouse Models of Cerebellar Ataxia. JoVE 2010, 1787, doi:10.3791/1787).

Moreover, we have used a combination of tests that allow us to rapidly quantify disease severity in a sensitive manner. We have incorporated various tests based on research on several disease models such as spinocerebellar ataxias, Huntington’s disease, and spinobulbar muscular atrophy. Our assessment includes hind limb clasping, ledge test, and kyphosis. Additionally, we have included gait analysis in our current evaluation, building upon our previous analysis.

Finally, as the ledge test allows the researcher to use any cage they have in possession, even the control mice may find the test more difficult depending on the cage used. Therefore, to address the reviewer’s comment, we have amended the scoring system for the ledge test, so that certain footslips, such as slips observed when the mouse is turning the corner of the ledge, would not automatically lead to a score of 1 or above. The revised scoring system reads as follows in 2.8 Behavioural analysis subsection of Methods [page 7, lines 253-263]:

“A score of 0 was assigned if the mouse generally did not lose its balance but exhibited non-consecutive slips while turning the corners of the ledge (corner slips). If the mouse experienced single-paw slips or transient crawling, a score of 1 was given. A score of 2 was assigned if the mouse had a total of one or two double-paw slips or major corner slips. Major corner slips were defined as slips where the mouse does not immediately regain its balance or, immediately after regaining balance, slips again. When a mouse had more than two double-paw slips or major corner slips, fell or nearly fell off the ledge, crawled for a prolonged period, or remained immobile for more than 20 seconds accompanied by tremor, it received a score of 3.” This method has now allowed to score more precisely the differences between mutants and controls and to reveal a progression of the deficit from 3- to 8M (new Figure 6A and B).

Comments 3: From the previous publication it is already known that the mutants have no deficit in the rotarod test. Taken together, the data appear unclear and also seem to be based on a very restricted sampling. Western blot analysis was done in 3 animals each, and the ledge test with 6 animals from each genotype. With these small group sizes, the chance of unreliable results is high.

Response 3: We regret any misunderstanding caused by the use of different cohort sizes in our behavioral tests. While the hindpaw clasping and kyphosis tests were carried out on smaller cohort sizes, the ledge test was carried out on 13 controls and 12 mutants at 3 months, and 9 controls and 16 mutants at 8 months. Therefore, we believe that these results are rather robust.

Comments 4: Based on presented data it is not justified to claim an ataxic phenotype. In addition, the presented cerebellar changes in the mutant mice could not explain the presence of such a phenotype. As the TrkB deletion is brain-wide in proenkephalin positive neurons it is not possible to assign the observed behavioral effects to the rather moderate cerebellar changes but it is more likely that they are due to effects in the other parts of the brain.

Response 4: We thank the reviewer for pointing out a possible limitation of our study. To address this, we have explored tdTomato expression in the cerebellar nuclei, brainstem and spinal cord of 3-month-old BAC-Penk-Cretg/+;Ai9 mice, as these parts of the central nervous system may affect motor behavior and consequently ledge test performance of TrkbPenk-KO mice. Only scattered recombination occurred in the cerebellar nuclei and the spinal cord of the TrkbPenk-KO mice, whereas in the brainstem, recombination occurred in the nuclei unrelated to motor behavior or ability to balance: cochlear nuclei, principal sensory and spinal nuclei of the trigeminal nerve, and the nucleus of the solitary tract. This information was included in 3.2 Pattern of BAC-Penk-Cre mediated recombination in the cerebellum subsection of Results [page 13, line 450; page 15, lines 491-501] and in Supplementary Figures 2 and 4.

It is known that recombination takes place in the striatum of TrkbPenk-KO. As we pointed out in the Discussion section, at 3 months, TrkbPenk-KO mice do not yet present with spontaneous hyperlocomotion, their muscle strength is normal and they are not impaired on the accelerated rotarod, suggesting that striatal dysfunction is unlikely to affect the ledge test performance of 3-month-old TrkbPenk-KO mice. However, we acknowledge that this remains a possible limitation, and we are currently carrying out pilot studies to eventually reduce any effects of striatal dysfunction on the motor behavior of TrkbPenk-KO mice.

Comments 5: Therefore, the authors are advised to do the following additional experiments:

1) Repeat the Western blot analysis with a new set of mice. Finally, there should be 3 independent experiments with mice from 3 different litters.

Response 5: We appreciate the reviewer’s concern over the cohort sizes in the Western blot analyses. Pursuant to this advice, we have used two cohorts of 6 animals each genotype at 3 months. For the Western blot analysis at 8 months, we included two additional cohorts of 6 animals (6 controls and 6 mutants total) in the comparisons of calbindin and synaptophysin levels. Moreover, we added an additional synaptic marker, PSD95, a postsynaptic density marker, to broaden our analysis of potential synaptic defects at 3- and 8-month stages together with GAD67. GAD67 and PSD95 levels were compared in one cohort of 6 animals (3 controls and 3 mutants) due to time limitations. In addition, we purchased a different synaptophysin antibody, which is highly regarded for its use in western blotting (Anti-Synaptophysin antibody [ab52636]). It is a rabbit recombinant monoclonal antibody, offering several advantages, including high batch-to-batch consistency and reproducibility. It also offers improved sensitivity and specificity.

The revised results were included in Figure 5 [page 17] and in the 3.4 Molecular consequences of Ntrk2 deletion in TrkbPenk-KO mice subsection of the Results [pages 17-18].

Due to the time limitations, we were not able to add another cohort of 6 at 3-month-old animals.

Comments 6: 2) Do a more extensive behavioral testing program. Rotarod is fine and has been done, add beam walking test and/or grid walking for better assessing the ataxic phenotype. The data presented so far are not convincing. Use a larger group of animals (approx. 20) for each genotype.

Response 6: We are grateful for the reviewer’s suggestions. To address the reviewer’s concern, we have carried out the CatWalk test on 3-5-month-old mice (5 controls, 9 mutants) to evaluate the gait of TrkbPenk-KO mice. Since 5-month-old mice TrkbPenk-KO mice have been tested on the CatWalk previously (Ref.17, Besusso et al., Nat Commun 2013) we restricted our analysis to the parameter groups previously shown to be affected by cerebellar ataxia in humans (Ref.39,  Buckley, E., Gait & posture60, 154-163, 2018) to focus on the cerebellar-induced effects. We found significant changes in two of the evaluated gait parameters: print positions of the left paws were significantly increased, whereas left hind swing duration was significantly reduced in TrkbPenk-KO mice. These results were included in 3.5 Behavioural consequences of Ntrk2 deletion in Enk-derived GCs subsection of Results [page 20, lines 616-627] and Figure 6(C-F) [page 19].

Comments 7: 3) Better assess Purkinje cell survival and morphology. Try to do a quantification of Purkinje cell number in some lobules.

Response 7: These are important questions raised by the reviewers. In order to answer them, we performed Purkinje cell counting in 4-month-old mice (3 controls, 3 mutants). We found that the total number of Purkinje cells, estimated as the sum of Purkinje cells in the simple lobule, lobules II and III, culmen and nodulus, was not significantly different between controls and mutants. Additionally, individual Purkinje cell counts specifically in lobule III, culmen and nodulus support the conclusion. This was included in subsection 3.3 Unaltered cerebellar morphology of TrkbPenk-KO mice of the Results [page 17, lines 538-543], Figure 4(H) [page 16] of the manuscript (see below).

Figure 4. (H) Total Purkinje cell count of 4M TrkbPenk-KO mice and controls did not differ, t(4)=0.345, p=0.748. Controls, n=3; mutants, n=3.

Supplementary Figure 7. Purkinje cell counts in selected lobules of TrkbPenk-KO cerebella. Scatter bar plot showing no significant differences in Purkinje cell counts of selected individual lobules at 4M between TrkbPenk-KO mice and controls. Lobule III, t(4)=1.220, p=0.290. Culmen: t(4)=0.546, p=0.614. Nodulus: t(4)=1.235, p=0.284. Controls, n=3, all females; mutants, n=3, all females. Values are means ± SEM. p statistic from unpaired, two-tailed, Student’s t-test.

Comments 8: 4) Explain the very high tdTomato expression in the meninges (Figs. 2 and 3).

Response 8: We thank the reviewer for raising this question. Since PENK staining of P8 and 3M BAC-Penk-Cretg/+;Ai9 cerebella did not stain the meninges at either stage (as seen from Figures 2 and 3 of the original manuscript), we believe that the very high tdTomato expression in the meninges likely reflect that meninges expressed Penk at an earlier stage of mouse development. In fact, there is evidence suggesting that Penk is expressed in meningeal fibroblasts during rat embryonal development, but its expression disappears shortly before birth (Hildebrand, B., et al.,1995, Neuropeptides29(2), 89-95; Hildebrand, B., et al.,1996, Naunyn-Schmiedeberg's archives of pharmacology354, 404-410). Whether meningeal fibroblasts are the cells with the very high tdTomato expression in the meninges of the BAC-Penk-Cretg/+;Ai9 mice is outside of the scope of this study.

Additionally, the brightness of tdTomato signal within a particular cell does not correlate with the amount of Penk expressed, rather the presence of tdTomato expression, no matter the amount, indicates that some Penk expression has occurred in the cell.

Comments 9: 5) The quality of the histological images in the pdf was very poor and no download of better-quality images was offered. This is most likely to too strong reduction during pdf production. This needs to be improved.

Response 9: We are grateful to the reviewer for pointing this out. We are providing better quality images with the re-submitted manuscript.

Comments 10: In addition, the manuscript would greatly benefit from doing the behavioral experiments in parallel with the mouse line with a specific deletion of TrkB in all cerebellar granule cells (Chen et al. 2011). These mice should be available to the authors as the senior author was a co-author of this publication. This would help to better assess the effects of the deletion in the pro-enkephalin positive granule cells vs. all granule cells.

Response 10: We acknowledge that the inclusion of mα6::Cre;TrkBfl/fl mice would benefit this study and broaden our conclusions. However, as we no longer have access to these mice, these experiments are outside the scope of the current study.

5. Additional clarifications

In addition to the revisions described in the responses to the reviewer, we have condensed subsections 3.2-3.4 of the Results section to improve the readability of the revised manuscript. Therefore, Figures 2 and 4 of the original manuscript were moved into Supplementary Files. Similarly, the assessment of the dendritic trees of Purkinje cells was moved to subsection 3.3 Unaltered cerebellar morphology of TrkbPenk-KO mice of the Results, as it became more appropriate after the addition of Purkinje cell counting to the subsection. Additionally, we converted the graphs presenting hindpaw clasping and kyphosis scores into one table (Table 3) in order to reduce the size of the revised Figure 6, which now includes the CatWalk results.

Round 2

Reviewer 1 Report

Comments and Suggestions for Authors

I believe the manuscript in its present form can be accepted for publication.

Reviewer 2 Report

Comments and Suggestions for Authors

The authors have taken substantial efforts in order to address the points raised by this reviewer and have done most of the additional experiments which were suggested. Although not all points have been fully addressed by the revisions and several open questions remain, this reviewer has no further major issues.